# Senescent preosteoclast secretome promotes metabolic syndrome associated osteoarthritis through cyclooxygenase 2

**Weiping Su[1,2†], Guanqiao Liu[1,3†], Bahram Mohajer[4†], Jiekang Wang[1], Alena Shen[5], Weixin Zhang[1], Bin Liu[1], Ali Guermazi[6], Peisong Gao[7], Xu Cao[1], Shadpour Demehri[4]\*, Mei Wan[1]\***

[1]Department of Orthopaedic Surgery, Russell H. Morgan Department of Radiology and Radiological Science, The Johns Hopkins University School of Medicine, Baltimore, United States; [2]Department of Orthopaedic Surgery, The Third Xiangya Hospital of Central South University, Changsha, China; [3]Division of Orthopaedics & Traumatology, Department of Orthopaedics, Southern Medical University Nanfang Hospital, Guangzhou, China; [4]Musculoskeletal Radiology, Russell H. Morgan Department of Radiology and Radiological Science, The Johns Hopkins University School of Medicine, Baltimore, United States; [5]University of Southern California, Dornsife College of Letters, Arts and Sciences, Los Angeles, United States; [6]Department of Radiology, Boston University School of Medicine, Boston, United States; [7]Johns Hopkins Asthma & Allergy Center, Johns Hopkins University School of Medicine, Baltimore, United States

**\*For correspondence:**
sdemehr1@jh.edu (SD);
mwan4@jhmi.edu (MW)

[†]These authors contributed equally to this work

## Abstract

**Background:** Metabolic syndrome–associated osteoarthritis (MetS-OA) is a distinct osteoarthritis phenotype defined by the coexistence of MetS or its individual components. Despite the high prevalence of MetS-OA, its pathogenic mechanisms are unclear. The aim of this study was to determine the role of cellular senescence in the development of MetS-OA.

**Methods:** Analysis of the human osteoarthritis initiative (OAI) dataset was conducted to investigate the MRI subchondral bone features of MetS-human OA participants. Joint phenotype and senescent cells were evaluated in two MetS-OA mouse models: high-fat diet (HFD)-challenged mice and STR/Ort mice. In addition, the molecular mechanisms by which preosteoclasts become senescent as well as how the senescent preosteoclasts impair subchondral bone microenvironment were characterized using *in vitro* preosteoclast culture system.

**Results:** Humans and mice with MetS are more likely to develop osteoarthritis-related subchondral bone alterations than those without MetS. MetS-OA mice exhibited a rapid increase in joint subchondral bone plate and trabecular thickness before articular cartilage degeneration. Subchondral preosteoclasts undergo senescence at the pre- or early-osteoarthritis stage and acquire a unique secretome to stimulate osteoblast differentiation and inhibit osteoclast differentiation. Antagonizing preosteoclast senescence markedly mitigates pathological subchondral alterations and osteoarthritis progression in MetS-OA mice. At the molecular level, preosteoclast secretome activates COX2-PGE2, resulting in stimulated differentiation of osteoblast progenitors for subchondral bone formation. Administration of a selective COX2 inhibitor attenuated subchondral bone alteration and osteoarthritis progression in MetS-OA mice. Longitudinal analyses of the human Osteoarthritis Initiative (OAI) cohort dataset also revealed that COX2 inhibitor use, relative to non-selective nonsteroidal antiinflammatory drug use, is associated with less progression of osteoarthritis and subchondral bone marrow lesion worsening in participants with MetS-OA.

**Conclusions:** Our findings suggest a central role of a senescent preosteoclast secretome-COX2/PGE2 axis in the pathogenesis of MetS-OA, in which selective COX2 inhibitors may have disease-modifying potential.

**Funding:** This work was supported by the National Institutes of Health grant R01AG068226 and R01AG072090 to MW, R01AR079620 to SD, and P01AG066603 to XC.

## Editor's evaluation

The manuscript presents novel findings that link, in mechanistic terms, metabolic syndrome with osteoarthritis. A central role of a senescent preosteoclast secretome-COX2/PGE2 axis has been established. The translational significance relates to the future use of selective COX2 inhibitors as disease-modifying agents in the osteoarthritis that accompanies metabolic syndrome.

## Introduction

During the past two decades, major advancements have been made in understanding the pathogenesis of osteoarthritis-the most common chronic articular disease associated with pain and disability. However, effective disease-modifying osteoarthritis therapies are still unavailable (*Misra et al., 2015*; *Veronese et al., 2017*; *Chen et al., 2020a*). The heterogeneous causes of osteoarthritis make developing such therapies challenging. Osteoarthritis has long been considered the consequence of a 'wear and tear' process that leads to cartilage degradation, which is initiated and/or accelerated by direct joint trauma and excessive mechanical overloading (*Heijink et al., 2012*; *Zhang and Jordan, 2010*). However, only 12% cases of symptomatic osteoarthritis are attributable to post-traumatic osteoarthritis (PTOA) of the hip, knee, or ankle (*Brown et al., 2006*), indicating that mechanisms other than biomechanical factors are involved in osteoarthritis development.

While the causal relationship between metabolic syndrome (MetS) and OA is still under debate, epidemiological and prospective clinical studies have clearly showed that osteoarthritis is strongly associated with metabolic diseases, including obesity, diabetes, dyslipidemia, and hypertension, which are the individual components of MetS (*Collins et al., 2018*; *Francisco et al., 2018*; *Misra et al., 2019*; *Mohajer et al., 2021*; *Chen et al., 2020b*). In particular, 59% of participants with osteoarthritis had MetS compared with 23% of the general population (*Puenpatom and Victor, 2009*). The middle-aged population with osteoarthritis has more than five times the risk of MetS compared with the age-matched population without osteoarthritis (*Puenpatom and Victor, 2009*). Participants with osteoarthritis have a higher prevalence of cardiovascular disease risk factors, including dyslipidemia, hypertension, and diabetes mellitus, independent of weight (*Saleh et al., 2007*; *Haara et al., 2003*; *Cerhan et al., 1995*; *Philbin et al., 1996*). Moreover, up to 81% of the elderly population have radiographic signs of hand osteoarthritis (*Banks and Lindau, 2013*), non-weight bearing joints closely associated with MetS. MetS-associated osteoarthritis (MetS-OA) is now considered a distinct osteoarthritis phenotype defined by the presence of MetS—both individual MetS components and MetS as a whole (*Zhuo et al., 2012*). Therefore, it is imperative to understand the pathogenic mechanisms for MetS-OA development and progression.

Rather than being a primarily cartilage-based disease, osteoarthritis involves changes in the subchondral bone microarchitecture that might precede articular cartilage damage (*Walsh et al., 2010*; *Suri et al., 2007*). Furthermore, increasing evidence suggests that pathological alterations in subchondral bone are not merely a secondary manifestation of osteoarthritis but are critical contributors to early osteoarthritis progression and its severity (*Su et al., 2020*; *Mazur et al., 2019*; *Burr, 1998*; *Burr and Gallant, 2012*; *Findlay and Atkins, 2014*; *Muratovic et al., 2019*). Subchondral bone includes trabecular bone and the subchondral bone plate, which is corticalized bone similar to that found in other locations. Human osteoarthritis studies using histological and imaging analyses showed that subchondral bone changes are generally characterized by increased bone sclerosis with thickening of the cortical plate, loss of subchondral trabecular rods with thickening of the remaining trabecular bone, alteration in subchondral bone three-dimensional morphology (*Haj-Mirzaian et al., 2018*), and formation of new bone at the joint margins (i.e. osteophytes) (*Li et al., 2013*). These changes were detected in late-stage human osteoarthritis. Subchondral bone changes during early-stage osteoarthritis remain unclear because of the lack of histological assessment of subchondral bone in human

osteoarthritis. Animal studies have found increased osteoclast number and activity, with a high bone turnover rate in early-stage PTOA mice, rats, and rabbits (*Kwan Tat et al., 2010*; *Zhen et al., 2013*). As osteoarthritis progressed in PTOA mice, increased formation of osteroid islets and trabecular irregularity were observed (*Kwan Tat et al., 2010*). Studies of the subchondral bone changes in MetS-OA are limited, especially during the early stage of this distinct osteoarthritis phenotype.

Cellular senescence has been viewed as a series of diverse and dynamic cellular states with irreversible cell-cycle arrest and the senescence-associated secretory phenotype (SASP) (*Campisi, 2013*; *Coppé et al., 2010*). Senescent cells (SnCs) exhibit stable cell-cycle arrest through the actions of tumor suppressors, such as *Cdkn2a, Cdkn2b, Trp53, Cdkn1a* (*Campisi, 2013*; *McHugh and Gil, 2018*). SnCs communicate with neighboring cells and influence the tissue microenvironment through SASP. Recently, it was found that SnCs increased in joint cartilage and synovium in mice after PTOA, and the selective elimination of SnCs led to attenuated osteoarthritis progression (*Jeon et al., 2017*). It remains unclear whether and how cellular senescence is involved in the pathogenesis of MetS-OA.

In the present study, we conducted analysis of the human osteoarthritis initiative (OAI) dataset to investigate the MRI subchondral bone features of MetS-human OA participants. We also characterized the joint phenotype of two MetS mouse models: high-fat diet (HFD)-challenged mice and STR/Ort mice. C57BL/6 mice fed a HFD are known to present key components of MetS relative to the mice fed a chow-food diet (CHD) (*Gallou-Kabani et al., 2007*) and to develop articular cartilage degeneration (*Sansone et al., 2019*). The STR/Ort mouse strain, an inbred substrain of STR/N mice (*Staines et al., 2017*), is a well-recognized model of spontaneous OA characterized by subchondral bone sclerosis, osteophyte formation, and articular cartilage degeneration. STR/Ort mice also develop hypercholesterolemia and hyperlipidemia (*Staines et al., 2017*; *Mason et al., 2001*), and therefore are a promising model for studying the pathogenic mechanisms of MetS-OA. We uncovered a unique, early, structural joint alteration of MetS-OA (i.e. subchondral bone thickening), which distinguishes MetS-OA from the joint changes of PTOA. We further elucidated a SASP-stimulated cyclooxygenase 2 (COX2)/prostaglandin E2 (PGE2) (COX2-PGE2) pathway that mediates the paracrine effect of senescent preosteoclasts on osteoblast lineage to promote subchondral bone formation. Finally, we examined the potential disease-modifying properties of selective COX2 inhibitors for MetS-OA in mice and humans.

## Materials and methods

### Mice and treatment

All experimental procedures were approved by and conducted in accordance with the Institutional Animal Care and Use Committee guidelines of The Johns Hopkins University. C57BL/6 J mice (stock no. 000664), CBA/J mice (stock no. 000656), and *ROSA26*[lsl-EYFP] mice (stock no. 006148) were purchased from The Jackson Laboratory (Bar Harbor, Maine). STR/Ort mice were purchased from Harlan Laboratories (Frederick, MD). *Cdkn2a*[flox/flox] mice were generated by Dr. Gloria H. Su's laboratory from the Department of Pathology, Columbia University Medical Center (*Qiu et al., 2011*). *Cdkn2a*[tdTom] reporter mice (C57BL/6 background) were generated by Dr. Norman E. Sharpless's laboratory from University of North Carolina School of Medicine (Chapel Hill, NC) (*Liu, 2019*). The *Tnfrsf11a*[Cre/+] mouse strain was generously provided by Yasuhiro Kobayashi (Matsumoto Dental University, Japan) (*Maeda et al., 2012*). *Tnfrsf11a*[Cre/+]; *ROSA26*[lsl-EYFP] mice (RANK-EYFP mice) were generated by crossing *Tnfrsf11a*[Cre/+] with *ROSA26*[lsl-EYFP] mice. We crossed the *Tnfrsf11a*[Cre/+] mice with *Cdkn2a*[flox/flox] mice. The offspring were intercrossed to generate *Tnfrsf11a*[Cre/+]; *Cdkn2a*[flox/flox] (p16[cko]) and *Cdkn2a*[flox/flox] (WT) mice. The genotypes of the mice were determined by PCR analyses of genomic DNA using the following primers: *Tnfrsf11a*[Cre/+] allele forward, 5'-GCAATCCCCAGAAATGCCAGATTAC-3'and reverse,5'-GCAAGAACCTGATGATGGACATGTTCAG-3'; *Cdkn2a*[flox/flox] allele (Note: *Cdkn2a* is the approved gene name for p16) P1, 5'-AGCAGCTTCTAATCCCAGCA-3' P2, 5'-CCACTCCTGGAACTCAGCAT-3' P3, 5'-AGGAGTCCTGGCCCTAGAAA-3' and P4, 5'-CCAAAGGCAAACTTCTCAGC-3'; *Cdkn2a*[tdTom] allele (Note: *Cdkn2a* is the approved gene name for p16) forward, 5'-ACCTCCCACAACGA-GGACTA-3' and reverse, 5'-CTTGTACAGCTCGTCCATGC-3'; *ROSA26*[lsl-EYFP] allele forward, 5'-AGGGCGAGGAGCTGTTCA-3' and reverse,5'-TGAAGTCGATGCCCTTCAG-3'.

Mice were housed in a 12 hr light/12 hr dark cycle with ad libitum water and food access. At 10–12 weeks of age, mice were placed on a Western HFD (21% fat by weight) (TD 88137, Harlan

Laboratories, Madison, WI) or a normal CHD for periods ranging from 2 weeks to 5 months. At the time of euthenasia, body weight was measured. The knee joints and serum were collected. For celecoxib treatment, mice were gavage-fed celecoxib at a dose of 16 mg/kg b.w for 2 months.

## Body composition and metabolic studies

Whole-body fat and lean body mass were assessed by quantitative nuclear magnetic resonance (echo MRI), as previously described (*Hu et al., 2020*; *Kim et al., 2019*; *Kim et al., 2017*). Plasma triglycerides (Sigma-Aldrich, St. Louis, MO), cholesterol (BioAssay Systems, Hayward, CA), and glycerol (Sigma-Aldrich) were measured colorimetrically. Glucose levels were measured using a hand-held OneTouch Ultra glucose monitor (LifeScan Inc, Milpitas, CA).

## MicroCT analysis

MicroCT analysis of the tibial subchondral bone was performed as previously described (*Su et al., 2020*; *Zhen et al., 2013*). The knee joint was analyzed by µCT (voltage, 65 kVp; current, 153 µA; resolution, 9 µm/pixel) (Skyscan 1174, Bruker MicroCT, Kontich, Belgium). The parameters of the tibia subchondral bone image were analyzed using reconstruction software (NRecon v1.6, Bruker), data analysis software (CTAn v1.9, Bruker), and three-dimensional model visualization software (µCTVol v2.0, Bruker). Three-dimensional structural parameters analyzed were BV/TV, Tb.Pf, Tb.Th, Tb.N, and SBP.Th. Ten consecutive images from the whole subchondral bone medial compartment were used to do three-dimensional reconstruction and analysis.

## Histology and Immunofluorescence staining

Mouse knee joints were harvested after euthanasia. For frozen sections, the bones were fixed in 4% formaldehyde overnight, decalcified in 1.5 M EDTA (PH = 7.4) for 14 days, and embedded in optimal cutting temperature compound. We used 30-µm-thick sagittal-oriented sections for immunofluorescent staining using a standard protocol. We incubated the sections with primary antibodies to mouse VPP3 (1:100, Abcam, Cambridge, UK), HMGB1 (1:500, Novus Biologicals, Littleton, CO), lamin B1 (1:100, Santa Cruz Biotechnology, Dallas, TX), F4/80 (Abcam, 1:50), OCN (1:200, Takara Bio Inc, Shiga, Japan) overnight at 4 °C followed by corresponding fluorescence-linked secondary antibodies (Jackson ImmunoResearch Laboratories, West Grove, PA) for 1 hr while avoiding light. The sections were then co-stained with 4',6-diamidino-2-phenylindole (H-1200, DAPI, Vector Laboratories, Burlingame, CA). The sample images were captured by a confocal microscope (Zeiss LSM 780). SA-βgal staining was conducted as previously described (*Li et al., 2017*; *Liu et al., 2021*), and the sample images were observed and captured by a microscope camera (DP71, Olympus BX51, Tokyo, Japan). For paraffin sections, bones were fixed in 4% formaldehyde overnight, decalcified in 1.5 M EDTA (PH = 7.4) for 21 days, and embedded in paraffin. We used 4-µm-thick sagittal-oriented sections for Safranin O (Sigma-Aldrich, S2255) and fast green staining (Sigma-Aldrich, 473 F7252). OARSI scores were calculated according to Safranin O–fast green staining. Osteoclasts were stained for TRAP. Osteoblasts were stained by OCN immunohistochemistry. Quantitative histomorphometry analyses were performed in a blinded fashion using OsteoMeasure Software (OsteoMetrics, Inc, Decatur, GA). Number of osteoblasts per bone marrow area (Ob.N/Bm. Ar.), osteocytes per bone area (Osteocytes N/B. Ar.), number of osteoclasts per bone perimeter (Oc.N/B. Pm) in the whole joint subchondral bone area per specimen. For each treatment group, 5–10 mice were used. For each sample, 3 tissue sections were used, and the whole joint subchondral bone area was analyzed.

## Cytokine array analysis of subchondral bone/bone marrow extracts

Knee joints from the mice were dissected, and tibia subchondral bone tissue was homogenized in lysis buffer containing 1% Triton X-100 and protease inhibitors. Protein extracts were collected and protein concentration were measured. The antibody array for secreted factors was performed using Mouse XL Cytokine Array Kit (ARY028, R&D Systems, Minneapolis, MN) according to the manufacturer's instructions.

## Quantitative real-time PCR

Total RNA for qRT-PCR was extracted from the cultured cells using RNeasy Mini Kit (74104, QIAGEN, Hilden, Germany) according to the manufacturer's protocol. For qRT-PCR, cDNA was prepared with

random primers using the SuperScript First-Strand Synthesis System (Invitrogen, Waltham, MA). Then qRT-PCR was performed with SYBR Green Master Mix (QIAGEN) using C1000 Thermal Cycler (Bio-Rad Laboratories, Hercules, CA). Relative expression was calculated by the $2^{-CT}$ method with GAPDH for normalization.

## CFU-F, CFU-Ob, and in vitro differentiation assays of mouse BMSCs

For CFU-F assays, freshly isolated single-cell suspensions from the long bone of 12-week-old male mice were plated in 6-well plates with a density of $5 \times 10^5$ cells per well. CFU-F colonies were counted after 10 days of culture with Crystal violet staining. We measured the colonies that had 50 cells or more. For CFU-Ob assays, freshly isolated bone marrow single-cell suspensions were seeded at a density of $5 \times 10^5$ cells per well in six-well plates. After 21 days of culture with StemPro Osteogenesis Differentiation Kits (Invitrogen), osteogenic differentiation was detected by Alizarin red staining. The colony-forming efficiency was determined by counting the number of colonies per $5 \times 10^5$ marrow cells plated.

## Generation of senescent preosteoclasts and preparation of CM

Preosteoclasts were generated in vitro as previously described (*Xie et al., 2014*). Briefly, monocytes/macrophages were isolated from the bone marrow of 3-month-old WT male mice by flushing cells from the bone marrow of femora and tibiae. The flushed bone marrow cells were cultured overnight on Petri dishes in α-MEM containing 10% fetal bovine serum, 1% penicillin-streptomycin solution, 40 ng/mL M-CSF (Amizona Scientific LLC, AM10003-600). The cells were then incubated with osteoclasto-genesis medium, containing αMEM with 10% fetal bovine serum, 1% penicillin-streptomycin solution, 40 ng/mL M-CSF and 60 ng/mL RANKL (Amizona Scientific LLC, AM10003-500) for 3 days, when most cells became mononuclear TRAP⁺ preosteoclasts (*Xie et al., 2014*). The formation of preosteoclasts was validated using TRAP staining (Sigma-Aldrich, 837) according to the manufacturer's protocol. The cells were then challenged with 80 µg/mL oxLDL (Alfa Aesar, Cat# J64164) or vehicle (control) for 24 hr. Cellular senescence was confirmed using SA-βgal staining, Lamin B1 immunocytochemical staining, and qRT-PCR of *Cdkn2a, Cdkn1a, Mki67* in the cells. CM was prepared by incubating the cells with serum-free medium for another 48 hr. After centrifugation (2500 rpm for 10 min at 4 °C), the CM was aliquoted and stored at –80 °C for different in vitro assays.

## RNA-sequencing analysis

Preosteoclasts were challenged with $H_2O_2$ (200 µM for 2 hr, then 20 µM for 1 day) or vehicle (control). Total RNA of pre-osteoclasts was isolated using TRIzol (Life Technologies, USA). RNA-Seq library construction and RNA high-throughput sequencing were performed. We analyzed aging/senescence-induced genes (ASIGs) from publicly available mouse RNA-seq data (Aging Atlas database; KEGG pathway database; GO database; MSigD database). GO, KEGG pathway, GSEA analysis were analyzed based on the comparison of our data with ASIGs.

## Statistical analysis of animal studies and in vitro cell culture studies

Data are presented as means ± standard errors of the mean. Unpaired, two-tailed Student *t*-tests were used for comparisons between 2 groups. For multiple comparisons, one-way analysis of variance with post hoc Tukey test was used. All data were normally distributed and had similar variation between groups. Statistical analysis was performed using SAS, version 9.3, software (SAS Institute Inc, Cary, NC). $p < 0.05$ was deemed significant. All representative images of bones or cells were selected from at least three independent experiments with similar results unless indicated differently in the figure legend.

## Human participant selection criteria

We selected OAI participants according to tailored stepwise criteria. (*Supplementary file 1D*) Knees with previous joint replacement surgery as shown in baseline radiographs were excluded (N=64, Exclusion #1, Supplementary Flowchart 1). From three OAI cohorts of knee osteoarthritis incidence, progression, and cohort non-exposed to osteoarthritis risk factors, we excluded participants from the non-exposed cohort of OAI (N=233, Exclusion #2, *Supplementary file 1D*) because of minimal risk of osteoarthritis incidence and progression. In order to assess OA-related subchondral BML damage,

we included participants with available BML MOAKS scoring on the baseline and follow-up MRIs. We collected and pooled all previously conducted MRI-based measurements of participants from nested ancillary studies performed inside OAI. These studies' design and selection criteria are specially tailored to assess MRI-based OA structural damage worsening in a specific subset from all OAI participants (details of these studies are explained in the OAI online repository) (*Overview and Description of Central Image Assessments, 2016*). Following deletion of duplicate measurements (753 cases between different projects), MRI Osteoarthritis Knee Score (MOAKS) measurements for 1671 knees were included from the following OAI ancillary studies: (1) Foundation for the National Institute of Health (FNIH) Consortium Osteoarthritis Biomarkers Project (473 knees, project no. 22), (2) project no. 30 (125 knees) (3) projects no. 63 A–63F (328 knees) (4) Pivotal OAI MRI Analyses (POMA) study (751 knees). The same OAI team centrally performed all measurements according to the validated semi-quantitative MOAKS (*Hunter et al., 2011*). Participants without available MRI reads were excluded (N=7614, Exclusion #3, *Supplementary file 1D*). To further homogenize our selection criteria and delineate the role of subchondral bone damage worsening in the MetS-OA, we included participants with MetS-OA without a history of knee trauma (to exclude PTOA cases) versus participants with PTOA without MetS (to exclude MetS-OA cases). MetS-OA participants with a history of knee trauma and PTOA participants with MetS were excluded (N=1041, Exclusion #4, *Supplementary file 1D*). These selection criteria are specifically aimed to investigate the unique pathophysiology (e.g. prominent subchondral bone damage) in the MetS-OA phenotype compared to PTOA, an OA phenotype with the central role of cartilage degeneration.

## Assessment of human knee osteoarthritis outcomes using Osteoarthritis Initiative (OAI) dataset

We conducted observational studies by analyzing the human OAI dataset, which consists of annual clinical and radiographic data for the 9,572 knees of 4796 participants' (*Peterfy et al., 2008*) with a follow-up of 8 years at annual time points (2004–2015, clinicaltrials.gov identifier: NCT00080171, https://nda.nih.gov/oai/). We selected 1671 participants who had available MRI Osteoarthritis Knee Score (MOAKS) scoring from OAI ancillary studies. We defined MetS presence and its components according to the International Diabetes Federation (IDF) criteria (*Alberti et al., 2006*). A history of the knee was assessed with the answer to the question "*have you ever injured your knee badly enough to limit your ability to walk for at least two days?*" and participants with a positive history of knee injury were considered as having PTOA. We further used a detailed selection criteria and propensity score (PS)-matching method (*Haj-Mirzaian et al., 2019*; *D'Agostino, 1998*).

## Definition of PTOA phenotype

Knees of OAI participants who responded positive to the question "ever injured badly enough your knee to limit the ability to walk for at least two days?" were regarded as having PTOA phenotype. To further homogenize selection criteria, we excluded participants with PTOA who had criteria for MetS (Exclusion #3, *Supplementary file 1D*).

## Definition of MetS-OA phenotype

No unified criteria for the definition of MetS exist; therefore, we defined MetS presence and components at the baseline visit according to the most widely accepted criteria defined by the IDF: (1) Hypertension was defined as systolic blood pressure (BP) of ≥130 mm Hg or diastolic BP of ≥85 mm Hg at baseline physical examination or as the use of BP-lowering medication indicated in the participants' medication inventory form. (2) Diabetes mellitus was indicated by self-report or the presence of oral or injectable anti-diabetic medications on the medication inventory form. (3) Dyslipidemia was defined as the use of lipid-lowering medications indicated in the participants' medication inventory form at baseline. (4) Abdominal obesity was defined as a waist circumference of ≥94 cm in men and ≥80 cm in women on physical examination. Participants with abdominal obesity and at least two of the three other components (dyslipidemia, diabetes mellitus, hypertension) were regarded as having MetS. Participants with MetS were regarded as MetS-OA phenotype, whereas participants who did not meet the IDF criteria for MetS were not. Participants with missing medication data on medication inventory forms were excluded (4 knees).

### Definition of COX2 inhibitor and non-selective NSAID use

At the baseline visit, participants who reported regular COX2 inhibitor use on the OAI medication inventory forms were categorized as COX2 inhibitor users, and the remaining participants were categorized as non-users. Among COX2 inhibitor users, participants with a≤1 year history of regular COX2 inhibitor use before the baseline time point were categorized as incident users (initiators). Participants with >1 year of regular COX2 inhibitor use were regarded as prevalent users and were excluded from the study (*Johnson et al., 2013*). Prevalent users of COX2 inhibitors were excluded from the study (150 knees), and only COX2 inhibitor incident users were included because the differences in duration of COX2 inhibitor use before participation in the study (1 month to 10 years) can lead to 'Neyman' bias (*Danaei et al., 2012*), in which very sick and very well participants are excluded from enrollment (*Danaei et al., 2012*). It is possible that chronic and long-lasting conditions associated with COX2 inhibitor use led to rapid deterioration in knee osteoarthritis, which resulted in exclusion from OAI enrollment. Use of cohort studies and new-exposure (i.e. 'incident use' of COX2 inhibitor) designs are recommended to avoid this bias (*Johnson et al., 2013*; *Danaei et al., 2012*; *Ray, 2003*).

### Treatment groups, per-protocol follow-up design

To assess COX2 inhibitor use, at each visit, participants reporting regular use of COX2 inhibitor during the last 12 months were coded as COX2 inhibitor users for that year, whereas negative reports of COX2 inhibitor use at each annual visit were considered as non-users for that year. Included participants were followed annually for 8 years (median, 6 years; interquartile range, 5 years). The same approach was used to assess non-selective NSAID use. Using propensity-score- (PS) matching method, COX2 inhibitor user were matched with non-selective NSAID users considering potential confounders. Following a 'per-protocol' design in our longitudinal analysis, which is associated with less selection bias in observational designs than in 'intention to treat' design (*Danaei et al., 2013*), participants who did not adhere to their assigned treatment (COX2 inhibitor vs. non-selective NSAID use) in each visit were right-censored from that visit (*Danaei et al., 2013*). Details of participant selection, PS-matching, and per-protocol design is explained in detail in Materials and methods section.

### Imputation

The pattern of missing data in covariates used in PS matching was assessed using the test of missing completely at random (Little's test), visual representation, and logistic regression models, which resulted in a 'missing not at random' pattern (*Sterne et al., 2009*) in the OAI dataset, with <1.5% of values missing for all matching variables. Despite the 'missing not at random' pattern of data, multiple imputation models were used according to previous studies to try to reduce possible bias (*Resseguier et al., 2011*).

### PS matching method

We matched study groups according to baseline characteristics to minimize confounding by indication bias. We first used PS matching to assess the presence of MetS on knee osteoarthritis outcomes. Next, we separately used PS matching to select participants for the two assessments of knee osteoarthritis and subchondral BML worsening outcomes in this study (1) comparison between MetS-OA and PTOA (2) comparison between COX2 inhibitor users and non-selective NSAID users. Possible confounders were investigated using a directed acyclic graph to assess causal inference (*Schisterman et al., 2009*). The selected covariates for PS matching were age (years), sex (male/female), body mass index (categorized in pentiles for ease of matching of kg/m$^2$), and race (Caucasian or non-Caucasian). For assessment of MetS presence, smoking (current smoker, yes/no), use of alcohol (<1 or≥1 unit/week), Physical Activity Score for the Elderly (PASE), baseline KL grade, and medial JSN OARSI grade were also included as covariates. Western Ontario and McMaster Universities Osteoarthritis Index (WOMAC) pain score and history of knee trauma were also included as covariates in the PS-matching model for COX2 inhibitor use. For every knee in the exposure group (presence of MetS or COX2 inhibitor use), one best-matched knee in the non-exposure group (PTOA or non-selective NSAID users, respectively) was selected according to the confounding above variables (1:1 match). We used the nearest-neighbor method with a caliper distance of 0.2 on the imputed dataset (using multiple imputation models, all variables had <1.5% missing). The best matches were defined as knees with the highest-level match on their PS, calculated using logistic regression. We calculated standardized

mean difference (SMD) to confirm the balanced outcome of matching between groups. Variables that did not optimally match in the PS matching method (SMD ≥0.1, baseline KL and JSN grades in the PS-matching for MetS vs. PTOA participants) were included in all subsequent statistical models as covariates of adjustment.

## Assessment of standard osteoarthritis outcomes

Longitudinal risk of knee osteoarthritis outcomes was assessed. Radiographic osteoarthritis incidence was defined as a Kellgren-Lawrence (KL) grade of ≥2 in follow-up assessment of knees with KL grades <2 at baseline (*Felson et al., 2011*). Progression was defined as an increase of 1 grade or more in medial Osteoarthritis Research Society International (OARSI) medial joint space narrowing (JSN) score ≥1 during follow-up assessments (*Felson et al., 2011*; *Reijman et al., 2007*). Knee radiographs were read centrally, and all KL grading and medial JSN OARSI scorings at baseline and follow-up points (1–8 years) are publicly available. For symptom assessment, participants' self-reported pain was assessed using Western Ontario and McMaster Universities Osteoarthritis Index (WOMAC) scores for pain and disability at baseline and follow-up points (1–9 years). WOMAC pain and disability scores were standardized to a range of 0–100 and were summed, resulting in a standardized combined WOMAC score. A standardized combined WOMAC pain/disability score of ≥80 in 2 consecutive years was used as the definition for non-acceptable symptom state (*Tubach et al., 2005*; *Gandek, 2015*; *Angst et al., 2005*) incidence. Time of standard osteoarthritis outcomes in the study was defined as the earliest year with each outcome (*Christensen, 1987*). For participants with no event, the last available follow-up was used as the time in the study (*Christensen, 1987*). We adjusted all analyses for participants' PS to minimize 'confounding by indication'.

## Assessment of subchondral BML structural damage worsening

Three Tesla MRI systems (Trio, Siemens Healthcare) were used for OAI MRI acquisition. Parameters and pulse sequence protocol of OAI MRIs have been previously reported (*Hunter et al., 2011*). For comparison of MetS-OA and PTOA participants (i.e. MetS-OA$^+$(PTOA$^-$) versus PTOA$^+$(MetS-OA$^-$)), we used available data on MOAKS scorings of the OAI ancillary studies. These include the available 1671 MOAKS measurements for the knees that we pooled from all previously conducted from nested ancillary studies performed inside OAI to assess OA-related subchondral bone damage (*Hunter et al., 2011*) [explained in detail in Materials and methods section and the OAI online repository (*Overview and Description of Central Image Assessments, 2016*)]. Next, to compare BML worsening between COX-2 inhibitor users and non-selective NSAID users, MOAKS measurements of the included participants were not available on OAI ancillary studies. Therefore, a musculoskeletal radiologist with 12 years of experience (S.D.) read and scored baseline and 24-month follow-up T2 knee MRIs, using the same validated MOAKS scoring method (*Hunter et al., 2011*). The reader was blind to the participants' group (COX2 inhibitor users versus non-selective NSAID users). A 24-month BML score worsening was defined as a whole- or within-grade change, where within-grade was defined as a definite visual change while not fulfilling a whole-grade change definition. Previously validated MOAKS BML measures (*Collins et al., 2016*) were used: (1) worsening in the number of affected subregions with BML (improvement, no change, worsening in 1 subregion, and worsening in ≥2 subregions), (2) maximum worsening in BML score (no change, within-grade worsening, worsening by 1 grade, and by ≥2 grades), (3) improvement in the number of affected subregions (yes/no), and (4) improvement in subregions BML score (yes/no) (*Collins et al., 2016*).

## Statistics

We used Cox proportional hazards regression to determine associations between either MetS presence or COX2 inhibitor use with outcomes. HRs and 95% CIs are reported. All modeling was conducted using complex sample analysis, in which matched users/non-users, and same-participant knees were included in the specific clusters with equal weight. Logistic mixed-effect regression models were used to assess 24 month worsening in the BML MOAKS scores while considering random intercept for each cluster of matched COX2 inhibitor: non-selective NSAID user and within-subject similarities (due to the inclusion of both knees of included knees). Analyses were performed using the R platform (version 4.3.3). Two-tailed p-values <0.05 were considered significant.

### Study approval

The animal protocol (MO20M127) was reviewed and approved by the Institutional Animal Care and Use Committee of The Johns Hopkins University. We used data from the longitudinal multi-center OAI study (2004–2015 clinicaltrials.gov identifier: NCT00080171). All 4796 enrolled patients gave written informed consent. Institutional review boards of four OAI collaborating centers have approved the OAI study's Health Insurance Portability and Accountability Act-compliant protocol (approval number: FWA00000068).

## Results

### Human participants with MetS-OA have a higher risk of subchondral bone marrow lesion worsening compared with those with PTOA

To delineate the specific pathophysiology of MetS-OA and further similarize our study sample with non-traumatic animal model models of MetS-OA, we excluded participants with MetS-OA who had a history of knee trauma (MetS+ PTOA−), and we selected participants with PTOA and without MetS as the control group (PTOA+ MetS−). A total of 630 knees met the inclusion criteria (explained in detail in Materials and methods section). After 1:1 PS matching, 338 matched knees were included, of which 169 were MetS-OA and 169 were PTOA. MetS-OA participants had a mean ± standard deviation (SD) age of 63.7±8.0 years and were 59.8% female. Similarly, PTOA participants had a mean ± SD age of 63.4±8.3 years and were 56.2% female. *Supplementary file 1A* shows the baseline characteristics of MetS+ and PTOA participants before and after matching. There was no imbalance in either of the

**Table 1.** Longitudinal comparison of the standard knee OA outcomes and subchondral BML worsening between human participants with MetS-OA and their matched PTOA participants (MetS+ PTOA− versus PTOA+ MetS−).

| | PS-matched MetS-OA versus PTOA participants |
|---|---|
| **Knee OA standard outcomes** | **Hazard ratio (95% Confidence Interval), p-value, Sample size, Number of events[*] [MetS-OA: PTOA]** |
| Knee OA incidence | 0.89 (0.56–1.41), p:0.609, N:184 (92:92), Event [34:36] |
| Knee OA progression | 1.04 (0.73–1.49), p:0.822, N:316 (158:158), Event [65:62] |
| Symptomatic incidence (NASS) | 0.95 (0.54–1.67), p:0.859, N:338 (169:169), Event [28:30] |
| **Subchondral BML Worsening (MOAKS)** | **Odds ratio (95% Confidence Interval), P-value, N:338 (169:169), Number of events[*] [MetS-OA: PTOA]** |
| Worsening in number of affected subregions with BML | 1.37 (1.06–1.77), p:0.015<br>Improvement, [16:31]<br>No change, [92:84]<br>Worsening, [61:54] |
| Worsening in the number of affected subregions | 0.44 (0.22–0.87), p:0.018<br>Yes, [31:16] |
| Maximum worsening in BML score | 1.17 (0.85–1.6), p:0.337<br>No change, [83:91]<br>Worsening by ≤1 grade, [59:54] by ≥2 grades, [27:24] |
| Improvement in the subregions' BML score | 0.82 (0.51–1.30), p:0.389<br>Yes, [90:98] |

Standard OA outcomes (baseline to 8th year) and validated MOAKS measures of subchondral BML worsening (between baseline and 24 month visit) were assessed between knees of participants with MetS-OA and their PS-matched knees of participants with PTOA. Cox proportional hazards were used for standard OA outcomes, and participants had a mean follow-up duration of 6.9 years (median and 1st and 3rd quartiles of 8 years). and logistic mixed-effect regression models were used for subchondral BML assessments. Knees of participants were matched for confounders using the 1:1 PS matching method. All analyses were adjusted for the baseline Kellgren-Lawrence (KL) and Osteoarthritis Research Society International medial joint space narrowing (OARSI JSN) grades of knees. Standard OA outcomes included knee OA incidence defined by KL grade ≥2 in participants with KL equal to 0–1, knee OA progression defined by partial or whole grade progression in OARSI JSN grade, and knee OA symptomatic incidence measured by NASS. Subchondral BML worsening was assessed using standard MOAKS measures. N corresponds to the total number of knees included in each analysis and the number of matched knees of MetS-OA and PTOA participants in the parenthesis.

[*] Number of events for each outcome has been shown separately for participants with MetS-OA and PTOA in the brackets.

BML: Bone marrow lesion, COX2I: Cyclooxygenase 2 inhibitor, MetS: metabolic syndrome, MOAKS: MRI Osteoarthritis Knee Score, NASS: non-acceptable symptomatic state, PS: propensity score, OA: osteoarthritis.

potential confounding variables included in the matching between MetS-OA and PTOA groups (standardized mean difference or SMD <0.1) except for their baseline KL and JSN grades (SMDs of 0.159 and 0.132) which both were included as covariates of adjustment in all further statistical analyses. The mean follow-up duration for standard knee OA outcomes assessment was 6.9 years (Median and 1st and 3rd quartiles of 8 years). Participants with MetS-OA, had similar risk of radiographic osteoarthritis incidence (radiographic hazard ratio [HR], 0.89; 95% CI, 0.56–1.41, and symptomatic HR, 0.95; 95% CI, 0.54–1.67) and progression (assessed by OARSI JSN, HR, 1.04; 95% confidence interval (95% CI), 0.73–1.49) compared to matched PTOA participants (*Table 1*). However, despite the similar risk of knee OA incidence and radiographic progression, participants with MetS-OA had increased odds of 24 month worsening in subchondral bone marrow lesions (BML) damage compared to participants with PTOA. This finding was evident as the higher odds of increased knee joint subregions with subchondral BMLs (odds ratio [OR], 1.37; 95% CI, 1.06–1.77). Despite MetS-OA participants display a similar risk of knee OA incidence and progression compared to PTOA participants, they have more subchondral bone damage worsening compared to the matched PTOA participants. Therefore, MetS-OA participants have unique subchondral bone damage worsening, distinguishable from that of PTOA.

## MetS mice develop rapid osteoarthritis-related subchondral bone changes in knee joints

To investigate subchondral bone changes during the progression of MetS-OA, we studied two MetS mouse models: high-fat diet (HFD)-challenged and STR/Ort mice. Mice fed a HFD for 3 months had higher body weight, body fat mass, and serum glucose relative to CHD mice (*Figure 1—figure supplement 1A-D*). Serum lipid oxidation products assessed by malondialdehyde (MDA) level were also markedly higher in the HFD mice than in the CHD mice (*Figure 1—figure supplement 1E*). Therefore, HFD mice developed key components of MetS. We investigated the changes in osteoarthritic joints of HFD mice. Three-month-old mice were placed on a CHD or HFD for various periods, and subchondral bone changes were assessed. Mice fed a HFD for 3 and 5 months had different degrees of cartilage degeneration, indicated by proteoglycan loss and OARSI scoring (*Figure 1A and B*). Cartilage degeneration was not found in mice fed a HFD for 0.5 month or 1 month. Notably, three-dimensional micro-computed tomography (μCT) analysis showed a high subchondral bone mass phenotype in HFD-challenged male mice (*Figure 1C* and *Figure 1—figure supplement 2*). Tibial subchondral bone volume (BV)/tissue volume (TV) ratio, subchondral bone plate thickness (SBP.Th), and subchondral trabecular bone thickness (Tb.Th) increased dramatically in mice fed a HFD (vs. CHD) for all time periods tested (*Figure 1D-F*), indicating an early and severe thickening of the subchondral plate and trabecular bone. Subchondral trabecular number (Tb.N) was reduced in mice fed a HFD for 1, 3, or 5 months (*Figure 1G*), likely because of the fusion of the trabeculae. HFD-challenged female mice had a same high subchondral bone mass phenotype (*Figure 1—figure supplement 3A*), with increased BV/TV, SBP.Th, and Tb.Th but decreased Tb.N (*Figure 1—figure supplement 3B-E*). Moreover, osteophyte formation, another characteristic of osteoarthritis (*Dieppe and Lohmander, 2005*), was also identified in mice fed a HFD for 5 months (*Figure 1—figure supplement 2*). However, unlike PTOA mice that have apparent trabecular bone irregularity, as indicated by an increase in trabecular pattern factor (Tb.Pf) (*Su et al., 2020*; *Zhen et al., 2013*), HFD mice had similar subchondral Tb.Pf relative to CHD mice (*Figure 1H*). Thus, HFD-challenged mice exhibited a unique subchondral bone phenotype distinguishable from that of PTOA mice.

We also investigated the joint changes in STR/Ort mice, which have been shown to develop osteoarthritis spontaneously early in life with concomitant hypercholesterolemia and hyperlipidemia (*Staines et al., 2017*; *Mason et al., 2001*). Male STR/Ort and CBA control mice were used because male mice are known to have a higher incidence of OA than female mice (*Staines et al., 2017*). Higher body weight (*Figure 1—figure supplement 4A*) and greatly increased serum triglyceride and cholesterol levels were detected in STR/Ort mice relative to those of CBA control mice (*Figure 1—figure supplement 4B, C*). Significant proteoglycan loss in joint cartilage (*Figure 1—figure supplement 4D*) and increased OARSI scores (*Figure 1—figure supplement 4E*) were detected only in 4-month-old mice but not in 2-month-old STR/Ort mice. However, an increase in subchondral bone mass was detected in STR/Ort mice as early as 2 months of age, with higher BV/TV ratio (*Figure 1—figure supplement 4F*), SBP.Th (*Figure 1—figure supplement 4G*), and Tb.Th (*Figure 1—figure supplement 4*), as well as unchanged Tb.N (*Figure 1—figure supplement 4J*). Collectively, the MetS-OA mice, similar to

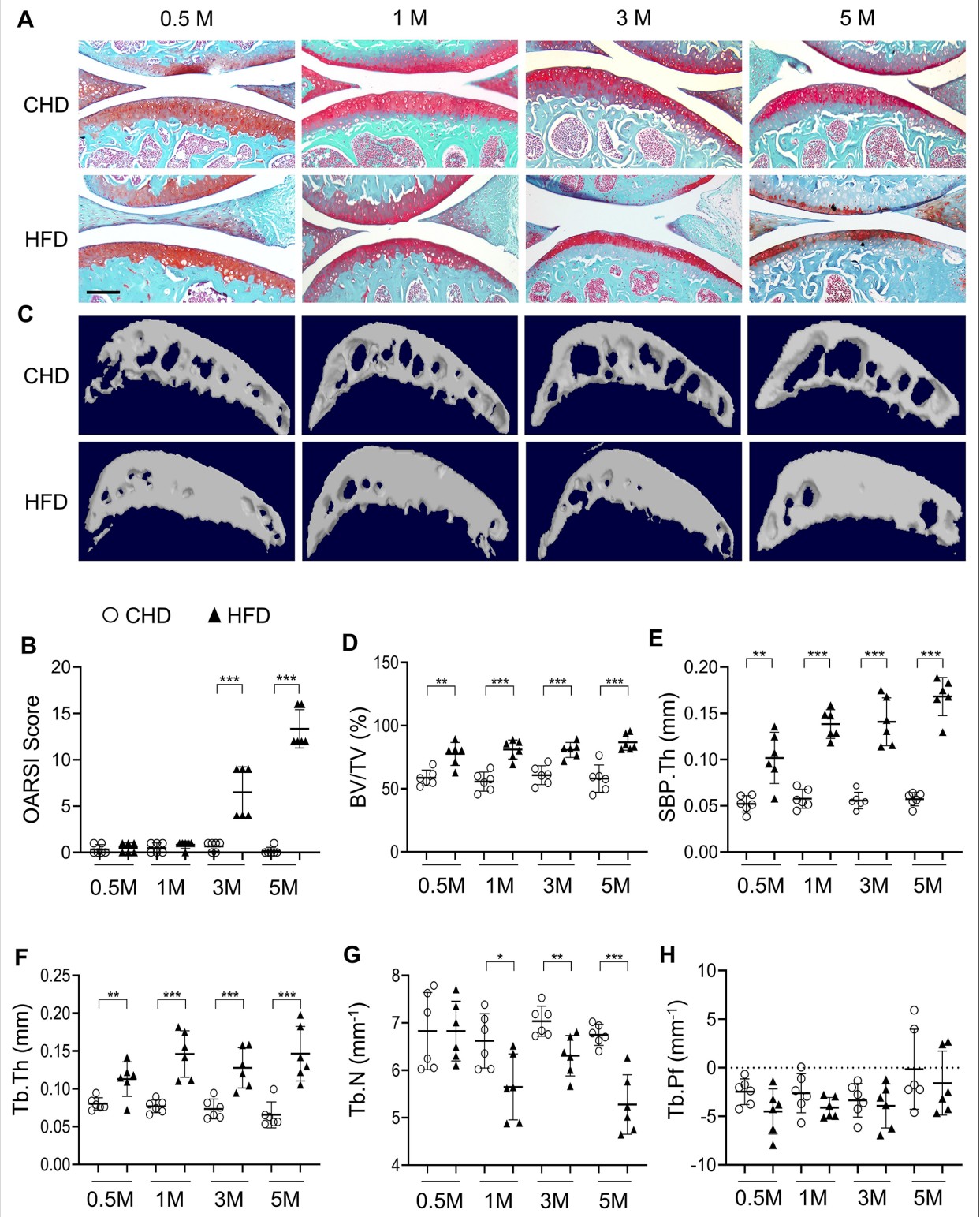

**Figure 1.** High-fat-diet (HFD) challenge leads to rapid subchondral bone thickening before cartilage damage occurs. Three-month-old C57BL/6 mice were fed a standard chow-food diet (CHD) or HFD for 0.5, 1, 3, or 5 months. n=6 mice per group. (**A**) Safranin O-fast green staining of the tibia subchondral bone medial compartment (sagittal view). Scale bar, 200 μm. (**B**) Calculation of Osteoarthritis Research Society International (OARSI) scores. (**C–H**) Three-dimensional micro-computed tomography (μCT) images (**C**) and quantitative analysis of structural parameters of knee joint subchondral bone: bone volume/tissue volume (BV/TV, %) (**D**), subchondral bone plate thickness (SBP. Th, mm) (**E**), trabecular thickness (Tb.Th, mm) (**F**), trabecular

*Figure 1 continued on next page*

*Figure 1 continued*

number (Tb.N, mm⁻¹) (**G**), and trabecular pattern factor (Tb. Pf, mm⁻¹) (**H**). All data are shown as means ± standard deviations. *p<0.05, **p<0.01, and ***p<0.001. Statistical significance was determined by unpaired, two-tailed Student's *t*-test.

The online version of this article includes the following figure supplement(s) for figure 1:

**Figure supplement 1.** Mice fed a high-fat diet (HFD) develop metabolic syndrome.

**Figure supplement 2.** Coronal and transverse view of micro-computed tomography scanning of joint subchondral bone change in high-fat-diet (HFD)-challenged mice.

**Figure supplement 3.** High-fat-diet (HFD) challenge leads to rapid subchondral bone thickening in female mice.

**Figure supplement 4.** STR/Ort mice develop dyslipidemia and subchondral bone thickness.

our findings in human MetS-OA participants, developed progressive subchondral bone damage and, importantly, exhibited rapid increases in subchondral bone plate and trabecular thickness before the occurrence of cartilage degeneration.

## Osteoblast and osteoclast lineage cells in subchondral bone change rapidly in response to HFD challenge

We then attempted to elucidate the cellular changes that underlie subchondral bone thickening in MetS. Immunostaining analysis showed that osteocalcin (OCN)⁺ osteoblasts were localized primarily on the subchondral bone surface in mice fed a CHD (**Figure 2A**). Although OCN⁺ cell number per bone surface was unchanged in HFD mice relative to CHD mice (**Figure 2B**), OCN⁺ cell number per bone marrow area markedly increased and formed clusters (**Figure 2A and C**) in mice at 0.5 month and 1 month after a HFD challenge. The results suggest that osteoblasts are located on subchondral bone surface in normal healthy joints but aberrantly accumulate in bone marrow cavity in the joints of MetS-OA mice. A higher number of osteocytes in subchondral bone was also detected in HFD mice relative to CHD mice (**Figure 2D**). Tartrate-resistant acid phosphatase (TRAP) staining showed reduced bone surface TRAP⁺ osteoclasts in HFD mice relative to CHD mice (**Figure 2E and F**). Consistently, HFD challenge induced a significant decrease in VPP3⁺ osteoclasts on the bone surface (**Figure 2G and H**). We studied *Tnfrsf11a*^Cre/+^; *ROSA26*^lsl-EYFP^ (RANK-EYFP) mice, in which RANK⁺ cells and their progeny are labeled with EYFP. Although the EYFP⁺ cells on the bone surface were reduced (**Figure 2I and J**), the EYFP⁺ cell number within the bone marrow cavity were elevated in HFD mice compared to those in CHD mice (**Figure 2I and K**). These results suggest that bone marrow RANK⁺ preosteoclasts may be unable to mature into bone surface osteoclasts.

## Subchondral preosteoclasts exhibit senescence-like phenotype in MetS mice

We reasoned that the reserved preosteoclasts in subchondral bone may undergo cellular senescence, causing an inability to migrate to the bone surface and fuse together into mature osteoclasts. We investigated SnCs in the joints of a senescence reporter transgenic mouse line, p16^tdTom^ (*Cdkn2a* is the approved gene name for p16 protein, and *Cdkn2a*^tdTom^ is used hereafter). Intriguingly, 1-month and 3-month HFD challenge led to the accumulation of tdTom⁺ SnCs exclusively in the subchondral bone (**Figure 3A and C**). tdTom⁺ SnCs were not seen in articular cartilage until later 5 months after HFD challenge (**Figure 3A and B**) when significant cartilage degeneration occurs (**Figure 1A, B**). Detection of other cellular senescence markers showed consistently increased SA-βGal⁺ cell number at the subchondral bone/bone marrow (**Figure 3D and E**) but not in articular cartilage in mice fed a HFD for 1 month or 3 months (**Figure 3D and F**). Conversely, fewer SA-βGal⁺ cells were found in the same region of CHD mice. We also performed immunostaining for HMGB1, the redistribution of which from nucleus to extracellular as a secretory protein initiates the process of cellular senescence (**Davalos et al., 2013**). Although most cells had strong nuclear staining of HMGB1 in CHD mice, many cells lost nuclear HMGB1 in the subchondral bone of the mice fed a HFD for 1 month or 3 months (**Figure 3—figure supplement 1A B**). Downregulation/loss of Lamin B1 has been recognized as a biomarker and crucial step for the development of cellular senescence (**Freund et al., 2012**). Unlike the well-preserved Lamin B1 expression in the cell membrane of CHD mice, many bone marrow cells lost Lamin B1 expression in the subchondral bone of HFD mice (**Figure 3—figure supplement 1C D**). We also investigated cellular senescence in the subchondral bone of STR/Ort mice. Similar to the

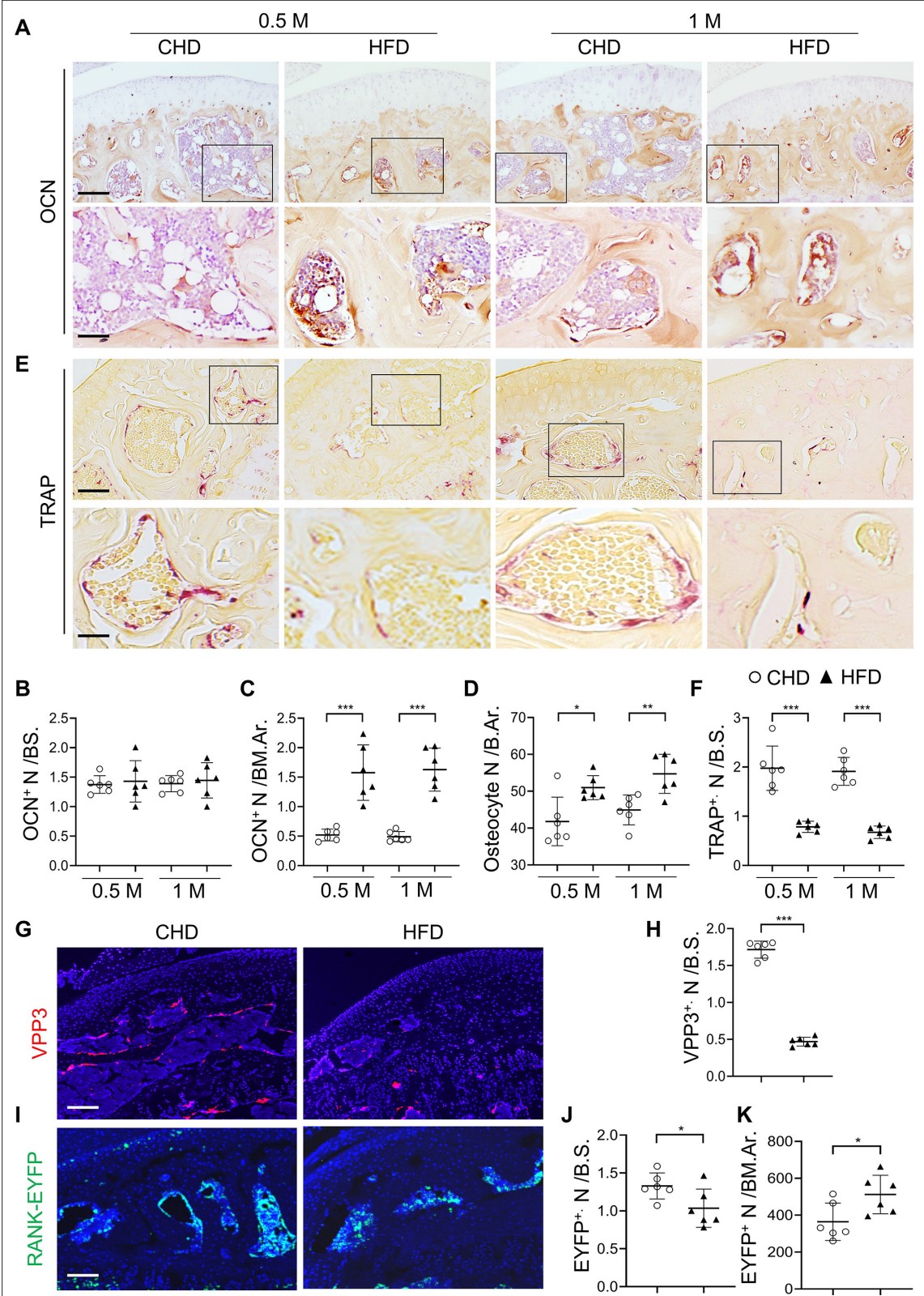

**Figure 2.** Subchondral osteoblast and osteoclast lineage cells change rapidly in response to a high-fat-diet (HFD) challenge. (**A–E**) Three-month-old C57BL/6 mice were fed a standard chow-food diet (CHD) or HFD for 0.5 month or 1 month. n=6 mice per group. Immunohistochemical staining of knee joint tissue sections with antibody against osteocalcin (OCN) (**A**). Quantification of OCN+ cells within bone marrow (BM) cavity (**B**) and on bone surface (BS) (**C**). Calculation of the number of osteocytes embedded in bone matrix (**D**). Scale bar, 100 μm(up), 50 μm(down). TRAP staining (**E**) and quantification

*Figure 2 continued on next page*

Figure 2 continued

of TRAP$^+$ cells at the bone surface (**F**). Scale bar, 100 µm(up), 50 µm(down). (**G–K**) Three-month-old *Tnfrsf11a$^{Cre/+}$; ROSA26$^{lsl-EYFP}$* mice were fed with CHD or HFD for 1 month. Immunofluorescence staining of knee joint tissue sections with antibody against VPP3 (red) (**G**). Quantification of the number of VPP3$^+$ cells at the bone surface (**H**). Scale bar, 100 µm. Immunofluorescence staining of knee joint tissue sections with antibody against GFP (green) (**I**). Quantification of the number of EYFP$^+$ cells at the bone surface (**J**) or in the BM area (**K**). Scale bar, 100 µm. Ar, area; BM., bone marrow. B.S., bone surface. All data are shown as means ± standard deviations. *p<0.05, **p<0.01, and ***p<0.001. Statistical significance was determined by unpaired, two-tailed Student's *t*-test.

HFD-challenged mice, the number of SA-βGal$^+$ cells in the subchondral bone marrow was higher in STR/Ort mice relative to CBA control mice at 2 months of age, when joint cartilage degeneration was not yet developed (**Figure 3—figure supplement 2A B**). Consistently, more subchondral bone marrow cells lost HMGB1 expression in STR/Ort mice relative to CBA mice (**Figure 3—figure supplement 2C D**), indicating that cellular senescence occurs at the pre- or early-osteoarthritis stage.

We then investigated whether osteoclast lineage cells undergo cellular senescence in HFD-challenged *Cdkn2a$^{tdTom}$* mice. Consistent with **Figure 2G–2H**, fewer VPP3$^+$ osteoclasts at the bone surface (**Figure 4A and B**) and more bone marrow tdTom$^+$ SnCs (**Figure 4A and C**) were detected in HFD mice than in CHD mice. However, the percentage of tdTom$^+$ cells out of the bone surface VPP3$^+$ cell population was not different in HFD mice compared with CHD mice (**Figure 4A and D**), indicating that mature osteoclasts did not undergo senescence in the subchondral bone of HFD-challenged mice. On the contrary, a markedly higher percentage of tdTom$^+$ SnCs in RANK$^+$ preosteoclasts was found in the subchondral bone marrow of HFD mice relative to CHD mice (**Figure 4E and F**). Approximately 49.2% and 56.5% of bone marrow RANK$^+$ cells expressed tdTom after HFD challenge of 0.5 and 1 month, respectively. These results suggest that bone marrow preosteoclasts, but not bone surface mature osteoclasts, exhibit senescence-like changes in mice with HFD challenge.

## Deletion of *Cdkn2a* in osteoclast lineage attenuates subchondral bone alterations and osteoarthritis progression in HFD mice

To determine whether cellular senescence plays a role in osteoarthritis development, we generated conditional *Tnfrsf11a$^{Cre/+}$; Cdkn2a$^{flox/flox}$* mice, in which the senescence gene *Cdkn2a* is deleted in RANK$^+$ osteoclast lineage cells (**Maeda et al., 2012**; **Zou et al., 2016**). Consistent with **Figure 3D**, SA-βGal$^+$ cells accumulated in subchondral bone marrow in WT mice with 1 month HFD treatment, and importantly this increase was greatly dampened in the p16$^{cKO}$ mice relative to WT mice (**Figure 5—figure supplement 1A B**). The results suggest that deletion of *Cdkn2a* in RANK$^+$ cells efficiently prevents/blocks subchondral cellular senescence. We then investigated subchondral bone changes in the mice. Consistent with **Figure 1**, µCT analysis showed that tibial subchondral BV/TV ratio, SBP.Th, and Tb.Th were all higher in wild-type (WT) mice fed a HFD (vs. CHD) (**Figure 5A-D**). However, these subchondral bone alterations induced by 1 month HFD treatment were not significant in the p16$^{cKO}$ mice. Moreover, the reduction in the number of bone surface osteoclasts (**Figure 5E and F**) and the increase in the osteoblast clusters in the bone marrow cavity (**Figure 5G and H**) induced by HFD treatment were both alleviated after *Cdkn2a* deletion in preosteoclasts. Of note, the µCT measurements (**Figure 5A-D**) and the histological parameters (**Figure 5E-H**) were not significantly different in the p16$^{cKO}$ mice compared with the WT mice fed a normal CHD, suggesting that subchondral bone remodeling, as well as the activity of osteoclast and osteoblast lineage were not affected by *Cdkn2a* deletion in osteoclast lineage cells at baseline. Finally, we investigated joint cartilage changes and found that articular cartilage was well preserved in WT and p16$^{cKO}$ mice fed a CHD. WT mice treated with 5 months of HFD exhibited obvious proteoglycan loss in the joint cartilage and an increased OARSI score, which were not observed in p16$^{cKO}$ mice (**Figure 5I and J**). Consistently, HFD-induced increase in the percentage of MMP13$^+$ chondrocytes, another feature of articular cartilage degeneration, was dramatically reduced in p16$^{cKO}$ mice relative to WT mice (**Figure 5K and L**). Therefore, deletion of *Cdkn2a* in osteoclast lineage cells attenuated HFD-induced osteoarthritis progression.

## Senescent preosteoclasts acquire a secretory phenotype

One of the characteristics of SnCs is the SASP. We investigated whether subchondral bone preosteoclasts in HFD mice acquire SASP by performing proteomic profiling. We assessed the differentially expressed proteins in tibial plateau subchondral bone from *p16$^{flox/flox}$* (WT) and p16$^{cKO}$ mice with HFD

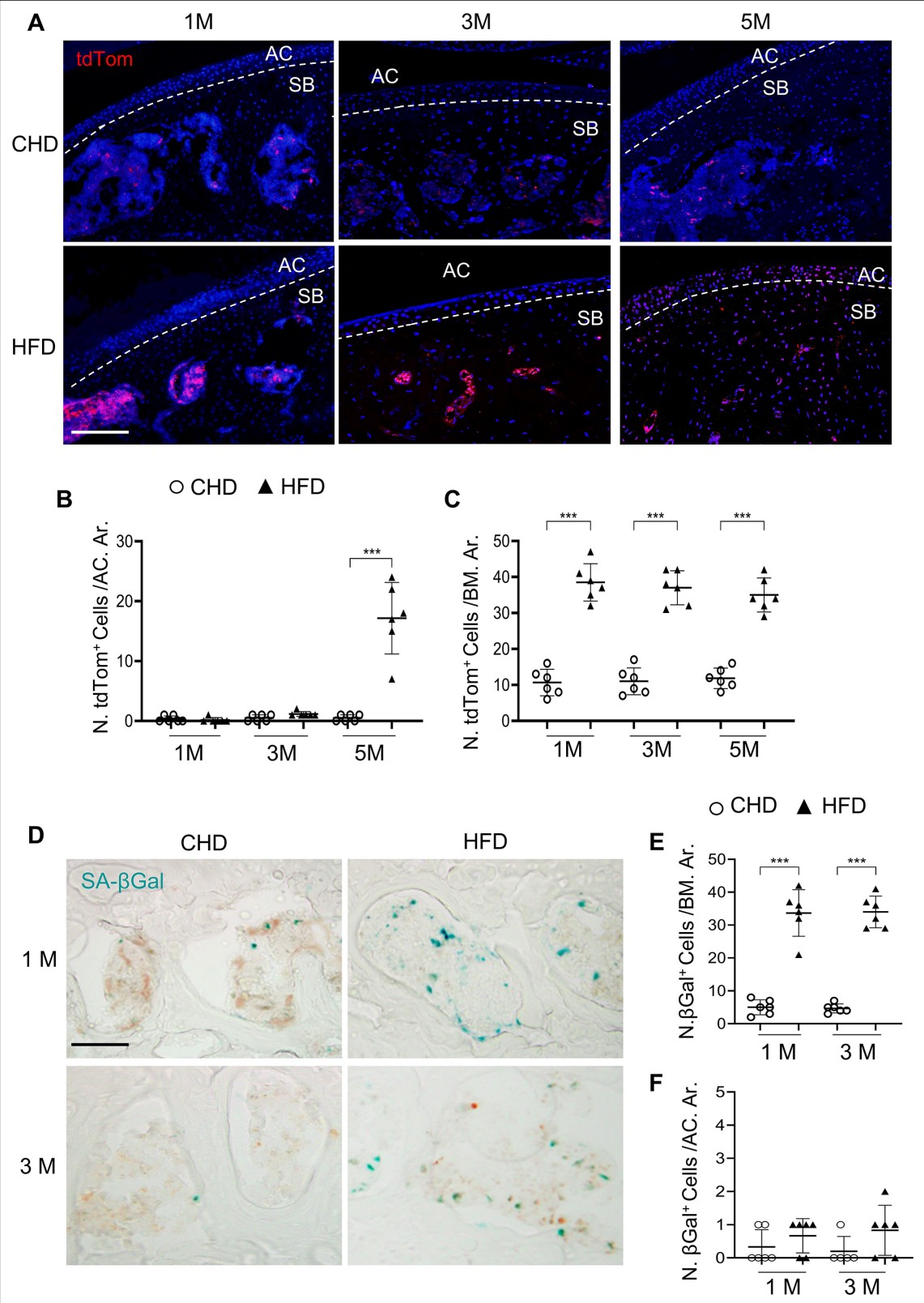

**Figure 3.** Senescent cells accumulate in subchondral bone of high-fat-diet (HFD)-challenged mice. (**A–B**) Three-month-old *Cdkn2a^tdTom* mice were fed a standard chow-food diet (CHD) or HFD for different time periods as indicated, n=6 mice per group. Fluorescence images showing tdTom+ cells (red) (**A**) with quantification of the number of tdTom+ cells at articular cartilage (**B**) or subchondral bone/bone marrow (**C**) of knee joints. Scale bar, 100 μm. (**D–**

*Figure 3 continued on next page*

*Figure 3 continued*

**F**) Three-month-old C57BL/6 mice were fed a CHD or HFD for 1 or 3 months. SA-βGal staining (**D**) and quantification of SA-βGal⁺ cells at subchondral bone/bone marrow (**E**) or articular cartilage (**F**) of knee joints. Scale bar, 100 μm.

The online version of this article includes the following figure supplement(s) for figure 3:

**Figure supplement 1.** Senescent cells accumulate in subchondral bone of high-fat-diet (HFD)-challenged mice.

**Figure supplement 2.** Senescent cells accumulate in subchondral bone of STR/Ort mice.

challenge (*Figure 6A*). Overall, 12 of 111 cytokines/growth factors were increased in the subchondral bone of HFD-challenged WT mice relative to that of those fed a normal CHD (*Figure 6B*). Of note, the levels of 9 of these 12 factors were restored in the subchondral bone of p16<sup>cKO</sup> mice, suggesting that these factors are secreted from the p16 +senescent preosteoclasts.

We then used an in vitro osteoclast-based cell culture system (*Xie et al., 2014*), in which isolated bone marrow monocytes/macrophages were treated with M-CSF and RANKL for different durations to obtain mononuclear preosteoclasts and multinuclear mature osteoclasts. Cellular senescence of the preosteoclasts was induced by challenging the cells with oxidized low-density lipoprotein (oxLDL) (*Figure 6C*), a key mediator of MetS-associated abnormalities in multiple tissues (*Holvoet et al., 2008*; *Hurtado-Roca et al., 2017*). The mRNA expression of senescence genes *Cdkn2a* and *Cdkn1a* was upregulated, and proliferation marker *Mki67* was downregulated in oxLDL-treated cells compared to those of vehicle-treated cells (*Figure 6D*). Moreover, number of SA-β-gal⁺ cells increased (*Figure 6E and F*) and the number of Lamin-B1⁺ cells decreased (*Figure 6G and H*) in response to oxLDL treatment. Therefore, oxLDL induces in vitro preosteoclast senescence efficiently. We then used quantitative real-time polymerase chain reaction (qRT-PCR) to evaluate whether the same secreted factors identified in the subchondral bone of HFD mice were also elevated in oxLDL-challenged preosteoclasts. oxLDL-treated cells had much higher expression of interleukin-1β (IL-1β) and IL-6, which are two common SASP factors. Moreover, seven factors identified in the subchondral bone of HFD mice, including osteopontin (OPN), Lipocalin-2, Cystatin C, IL-33, vascular endothelial growth factor (VEGF), cellular communication network factor 4 (CCN4), and platelet-derived growth factor BB (PDGF-BB), were also markedly increased in oxLDL-treated cells relative to vehicle-treated cells (*Figure 6I*), indicating that these are important SASP factors involved in osteoarthritis pathogenesis.

### Senescent preosteoclasts have diminished differentiation capacity and secrete SASP to inhibit differentiation of non-senescent osteoclast precursors

We tested whether the factors secreted by senescent preosteoclasts act on surrounding cells to regulate their activity in a paracrine manner. To do this, we prepared conditioned medium (CM) from control non-senescent preosteoclasts (Con-CM) and senescent preosteoclasts (SnC-CM) (*Figure 6C*). We first examined the functional changes of osteoclast lineage cells by incubating bone marrow Mo/Mac, osteoclast precursor cells, with the SnC-CM. SnC-CM inhibited the osteoclastogenesis ability of the non-adherent bone marrow Mo/Mac compared with Con-CM (*Figure 6—figure supplement 1*), indicating an inhibitory effect of preosteoclast SASP on osteoclastogenesis. We then test whether preosteoclasts lose their capacity to further differentiate into mature osteoclasts when become senescent. Cellular senescence of in vitro differentiated mononuclear preosteoclasts was induced by challenging the cells with hydrogen peroxide (H₂O₂) (*Figure 6—figure supplement 2A*), which has been widely used to achieve oxidative stress-induced cellular senescence (*Toussaint et al., 2000*). Senescence of preosteoclasts was successfully induced as indicated by upregulated expression of senescence genes *Cdkn2a* (*Figure 6—figure supplement 2B*) and nuclear loss of HMGB1 (*Hernandez-Segura et al., 2018*; *Figure 6—figure supplement 2C*) relative to vehicle-treated control cells. Moreover, bulk RNA-seq analysis revealed 4,056 differentially expressed genes in the senescent vs. control preosteoclasts (p<0.05). Comparison of our data with previously defined aging/senescence-induced genes (ASIGs) from publicly available mouse RNA-seq data sets (Aging Atlas database; KEGG pathway database; GO database; MSigD database) identified a total of 150 ASIGs in the senescent preosteoclasts (vs. control non-senescent preosteoclasts) (*Figure 6—figure supplement 3A*). Among these ASIGs, 31 genes were upregulated, and 119 genes were downregulated in the senescent preosteoclasts relative to control cells (*Figure 6—figure supplement 3B*). In the main biological process and molecular

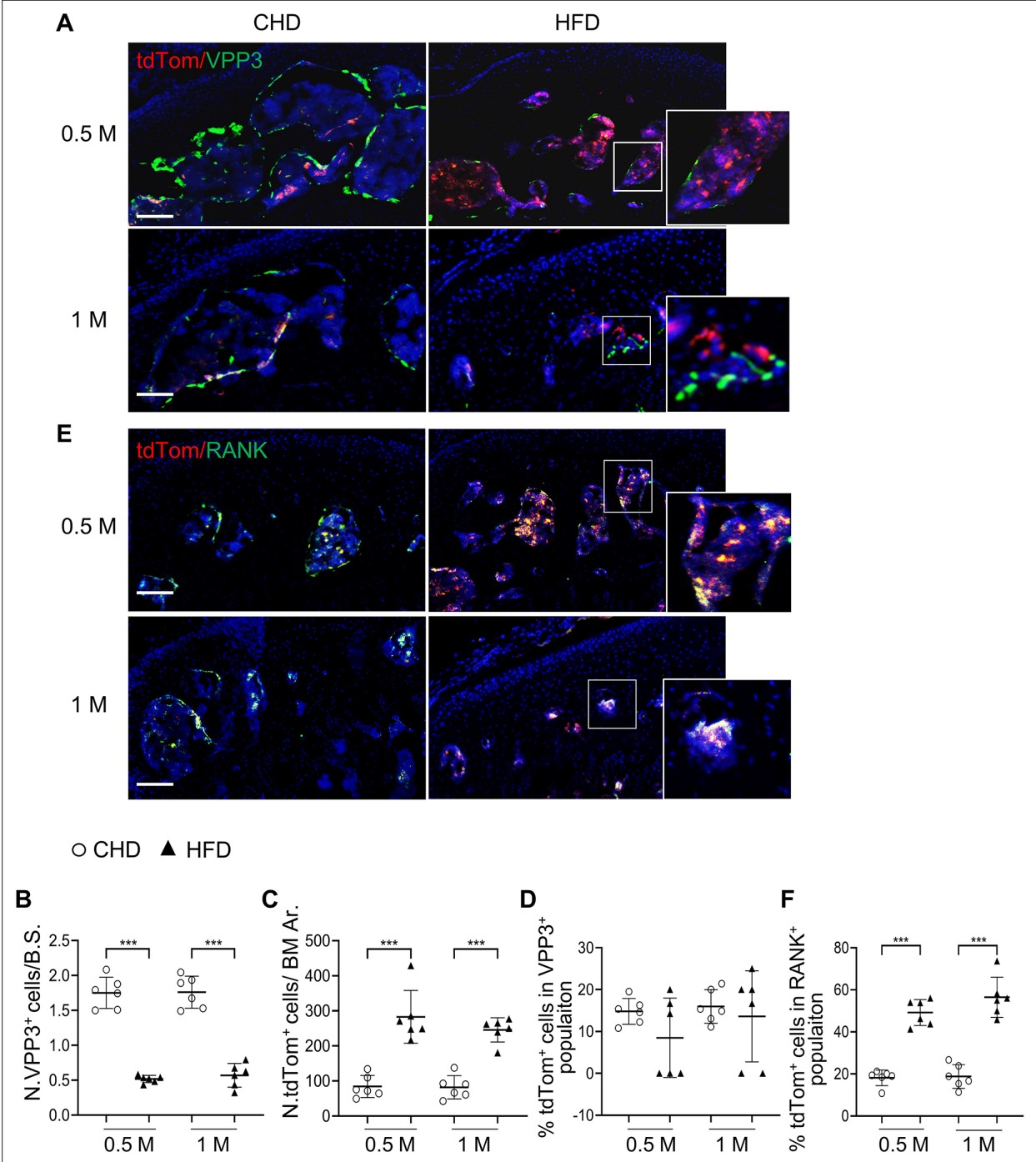

**Figure 4.** Subchondral marrow preosteoclasts exhibit senescence-like feature in high-fat-diet (HFD)-challenged mice. Three-month-old *Cdkn2a^tdTom* mice were fed a CHD or HFD for 0.5 month or 1 month, n=6 mice per group. Immunofluorescence staining of knee joint tissue sections with antibody against VPP3. Double fluorescence imaging of tdTom (red) and VPP3 (green) are shown in (**A**). Quantification of VPP3+ cell numbers at the bone surface (**B**); tdTom+ cell numbers in subchondral bone marrow area (**C**); percentage of tdTom+ cells out of total VPP3+ cells (**D**). Immunofluorescence staining of knee joint tissue sections with antibody against RANK. Double fluorescence imaging of tdTom (red) and RANK (green) were shown in (**E**). Percentage of tdTom+ cells out of total RANK+ cells were shown in (**F**). Scale bar, 100 μm. All data are shown as means ± standard deviations. ***p<0.001. Statistical significance was determined by unpaired, two-tailed Student's *t* test.

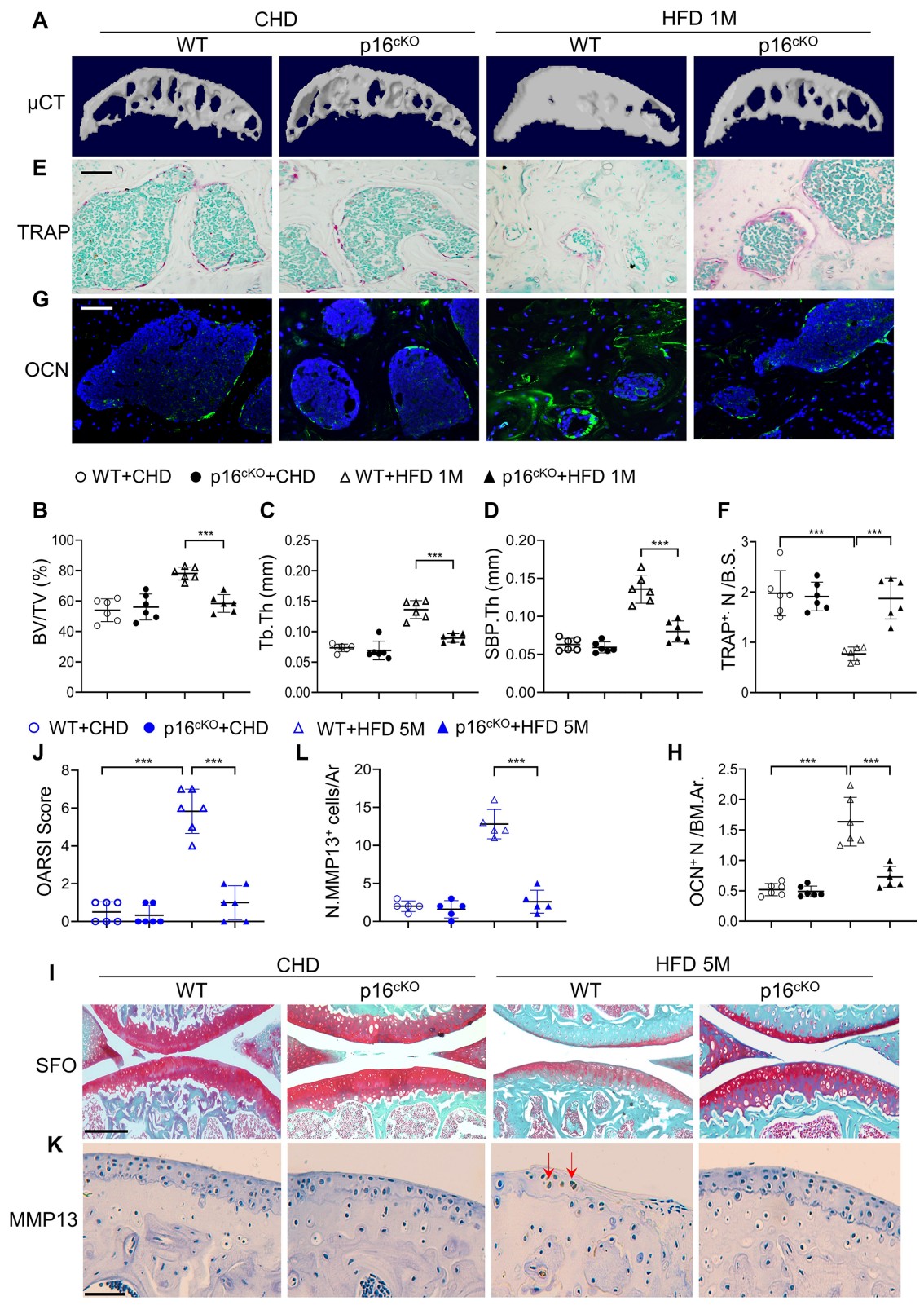

**Figure 5.** Deletion of *Cdkn2a* in preosteoclasts attenuates subchondral bone thickness and cartilage damage in high-fat-diet (HFD) mice. (**A–H**) Three-month-old *Tnfrsf11a^Cre/+^; Cdkn2a^flox/flox^* mice (p16^cKO^) and *Cdkn2a^flox/flox^* littermates (wild-type [WT]) were fed a standard chow-food diet (CHD) or HFD for 1 months, n=6 mice per group. Three-dimensional micro-computed tomography (µCT) images (**A**) and quantitative analysis of structural parameters of knee joint subchondral bone: bone volume/tissue volume (BV/TV, %) (**B**), subchondral bone plate thickness (SBP. Th, mm) (**C**), and trabecular

*Figure 5 continued on next page*

*Figure 5 continued*

thickness (Tb.Th, mm) (**D**). TRAP staining of knee joint tissue sections (**E**) and quantification of TRAP$^+$ cells at the bone surface (**F**). Scale bar, 100 µm. Immunofluorescence staining of knee joint tissue sections with antibody against osteocalcin (OCN) (green) (**G**). Quantification of the number of OCN$^+$ cells per bone marrow area (**H**). (**I–L**) Three-month-old *Tnfrsf11a$^{Cre/+}$; Cdkn2a$^{flox/flox}$* mice (p16$^{cko}$) and *Cdkn2a$^{flox/flox}$* littermates (wild-type [WT]) were fed a standard chow-food diet (CHD) or HFD for 5 months, n=6 mice per group. Safranin O-fast green staining of the tibia subchondral bone medial compartment (sagittal view) (**I**). Scale bar, 200 µm. Calculation of Osteoarthritis Research Society International (OARSI) scores (**J**). Scale bar, 100 µm. Immunostaining of knee joint tissue sections with antibody against MMP13 (brown) (**K**) and quantification of MMP13$^+$ cells on cartilage (**L**). Red arrows: MMP13$^+$ cells. Results are expressed as mean ± standard deviations, ***p<0.001. Statistical significance was determined by unpaired, two-tailed Student's *t* test.

The online version of this article includes the following figure supplement(s) for figure 5:

**Figure supplement 1.** Deletion of *Cdkn2a* in RANK$^+$ cells efficiently prevents subchondral cellular senescence.

function genes, alterations in those involved in 'Aging', 'Damaged DNA binding', 'chromatin DNA binding', and 'NF-kappaB binding' were notable for their known links to senescence and SASP triggering (*Tilstra et al., 2012*; *Di Micco et al., 2021*; *Figure 6—figure supplement 3C*). Of note, osteoclast differentiation- and bone resorption-associated genes are among the most significantly downregulated genes (*Figure 6—figure supplement 4*), indicating a diminished osteoclast differentiation capacity of the preosteoclasts after becoming senescent. Therefore, the senescent preosteoclasts, on one hand, have declined differentiation capacity toward mature osteoclasts; on the other hand, secrete SASP factors that inhibit the differentiation of non-senescent osteoclast precursors to osteoclasts in a paracrine manner.

## Preosteoclast secretome promotes osteoblast differentiation via COX2-PGE2 signaling

We then tested whether SnC-CM affects the activity of osteoblast lineage cells. Human bone marrow stromal cells (BMSCs), precursors of osteoblasts, incubated with SnC-CM, had greater osteoblast differentiation capacity than Con-CM, as detected by colony forming unit–osteoblasts (CFU-OB) (*Figure 7C and D*). SnC-CM did not affect the colony-forming capacity of the BMSCs (*Figure 7A and B*). Consistently, the expression of osteoblast differentiation markers alkaline phosphatase (*Alpl*), *Bglap*, and collagen type 1 A (*Col1a1*) were all upregulated in BMSCs incubated with SnC-CM (vs. Con-CM) (*Figure 7E*). The expression of RUNX2, the master regulator for the commitment of undifferentiated mesenchymal stem cells toward the osteoblast lineage, was unchanged in the cells treated with SnC-CM compared with Con-CM. These results suggest that the SASP factors produced by preosteoclasts promote late-stage osteoblast differentiation without affecting the lineage commitment of BMSCs.

We explored the mechanisms by which preosteoclast-secreted factors affect osteoblast differentiation. Most of these SASP factors, such as IL-1β, IL-6, VEGF, PDGF-BB, OPN, Lipocalin-2, myeloperoxidase (MPO), Cystatin C, resistin, and IL-33, are direct COX2 gene-activating factors (*Hurtado-Roca et al., 2017*; *Chien et al., 2009*; *Zhang et al., 2010*; *Su et al., 2017*; *Li et al., 2018*; *Hamzic et al., 2013*; *Panagopoulos et al., 2017*; *Samad et al., 2001*). COX2 is the inducible enzyme in the production of PGE2, which acts on osteoblastic precursors to stimulate osteoblast differentiation (*Pilbeam, 2020*). We detected the mRNA expression of COX2 in BMSCs using various culture media and measured PGE2 produced by the cells. COX2 gene expression was upregulated in the cells incubated in osteoblast differentiation medium (DM) compared with those incubated in growth medium (GM). Importantly, the addition of SnC-CM, but not Con-CM, stimulated much higher COX2 expression in the cells compared to those incubated with DM alone (*Figure 7F*). PGE2 protein level was also higher in the culture medium from the cells incubated with SnC-CM relative to that of the cells incubated with DM alone (*Figure 7G*). Further, treating the cells with selective COX2 inhibitor celecoxib significantly downregulated all 3 osteoblast differentiation marker genes stimulated by SnC-CM (*Figure 7H*), suggesting a requirement of the COX2-PGE2 pathway in the osteoblast differentiation of the BMSCs. Our results suggest that preosteoclast-derived SASP factors have a paracrine effect on surrounding osteoblast lineage cells for COX2-PGE2 activation, which consequently promotes osteoblast differentiation in a cell-autonomous fashion (*Figure 7I*).

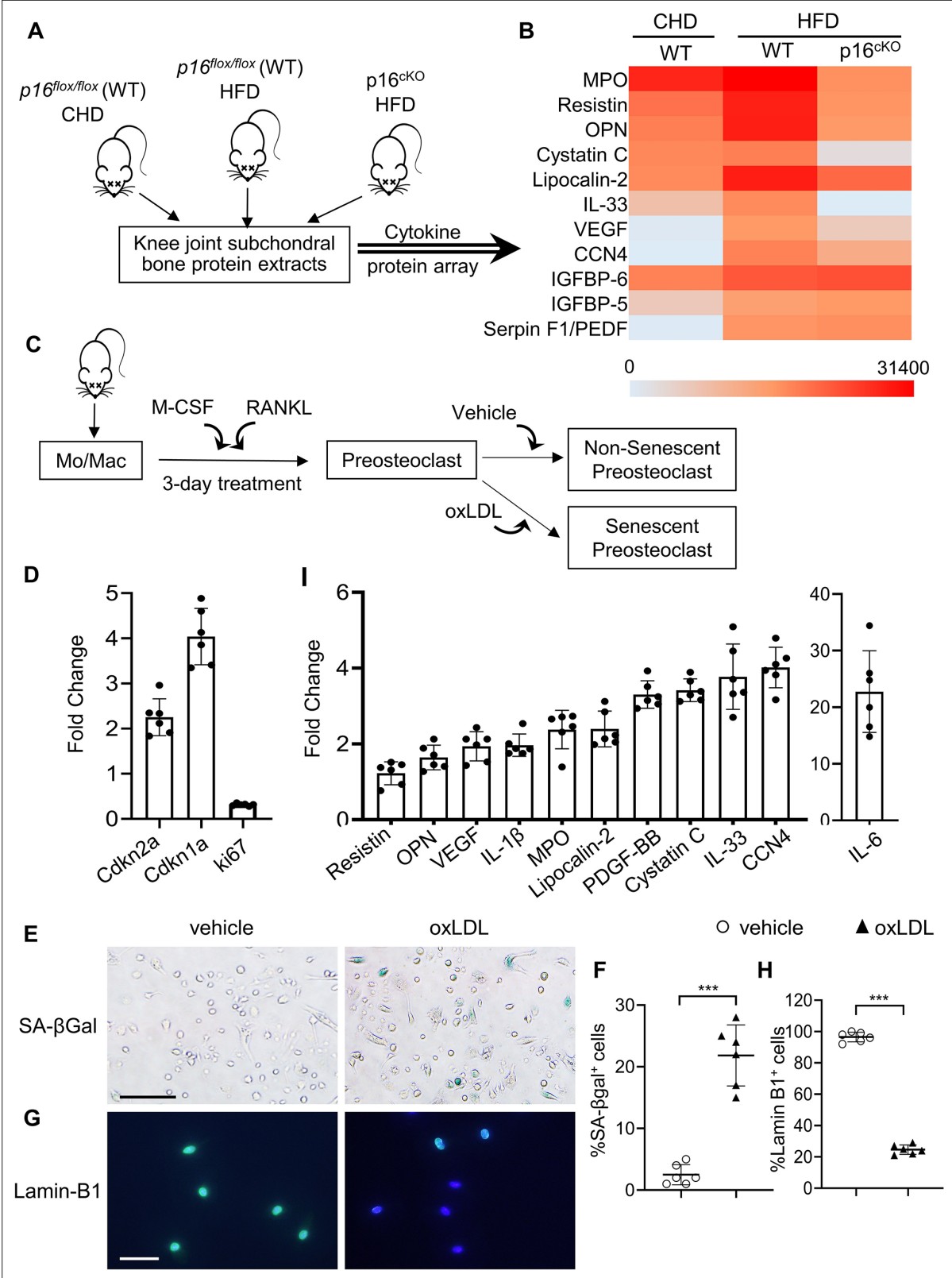

**Figure 6.** Subchondral preosteoclasts acquire a unique secretory phenotype in high-fat-diet (HFD) mice. Three-month-old *Cdkn2a^flox/flox* mice were fed a standard chow-food diet (CHD). Three-month-old *Tnfrsf11a^Cre/+; Cdkn2a^flox/flox* mice(p16^cko) and *Cdkn2a^flox/flox* littermates (wild-type [WT]) were fed with the HFD. (**A**) The subchondral bone protein extracts were harvested after 1 month. n=6 mice per group. (**B**) Differentially expressed proteins are shown on a heat map. (**C**) Isolated bone marrow monocytes/macrophages were treated with M-CSF and receptor activator of nuclear factor kappa-B ligand (RANKL)

*Figure 6 continued on next page*

*Figure 6 continued*

to obtain mononuclear preosteoclasts. Cellular senescence of the preosteoclasts was induced by challenging the cells with oxidized low-density lipoprotein (oxLDL) or vehicle. (**D**) In vitro senescence-associated changes in normalized mRNA expression of three senescence effectors. (*Cdkn2a*, *Cdkn1a*, and *Mki67*) (**E, F**) SA-βGal staining (**E**) and percentage quantification of SA-βGal⁺ cells (**F**). Scale bar, 100 μm. (**G–H**) Immunofluorescence staining of Lamin B1 (green) (**G**) and percentage quantification of Lamin B1⁺ cells (**H**). Scale bar, 50 μm. (**I**) In vitro senescence-associated changes in normalized mRNA expression of established SASP components are shown. Results are expressed as mean ± standard deviations, ***p<0.001. Statistical significance was determined by unpaired, two-tailed Student's *t*-test.

The online version of this article includes the following figure supplement(s) for figure 6:

**Figure supplement 1.** Senescence-associated secretome inhibits osteoclastogenesis.

**Figure supplement 2.** Senescence of preosteoclasts was successfully induced with hydrogen peroxide.

**Figure supplement 3.** RNA-seq reveals aging/senescence-induced genes in senescent preosteoclasts.

**Figure supplement 4.** Osteoclast differentiation-related genes are down-regulated in the senescent preosteoclasts in the RNA-seq dataset.

## Selective COX2 inhibitor attenuates HFD-induced subchondral bone changes and osteoarthritis progression in mice

To further define the role of COX2-PGE2 pathway in stimulating subchondral bone formation in vivo, we assessed COX2 expression in the subchondral bone of HFD-challenged mice using immunostaining analysis. Our results showed that there were very few COX2⁺ cells at cartilage in mice fed HFD for shorter periods (0.5, 1, 3, and 4 months) relative to mice fed CHD, whereas there was a much higher number of COX2⁺ cell number on bone surface (osteoblasts) and in mineralized bone (osteocytes) at these time points (*Figure 8A, C and D*). A higher number of COX2⁺ chondrocytes in cartilage was found in mice only after 5 months of HFD treatment (*Figure 8A and B*), during which severe cartilage degeneration occurs. Therefore, the majority of COX2⁺ cells were primarily subchondral osteoblasts and osteocytes during the progression of MetS-OA.

To investigate whether inhibiting COX2-PGE2 signaling could alleviate subchondral bone changes, such as subchondral plate thickening, and thus slow osteoarthritis progression in MetS-OA mice, we gavage-fed a selective COX2 inhibitor, celecoxib, to HFD mice at a dose of 16 mg/kg⁻¹ daily for 2 months. Subchondral bone alterations induced by the HFD were alleviated by celecoxib treatment relative to vehicle treatment (*Figure 9A*), as evidenced by significantly reduced subchondral BV/TV, SBP.Th, and Tb.Th. (*Figure 9B-D*). The reduction of Tb.N induced by a HFD was also normalized by celecoxib treatment (*Figure 9E*). Importantly, the degeneration of articular cartilage was also ameliorated after celecoxib treatment (*Figure 9F and G*). Celecoxib treatment in CHD mice did not change the architecture of the subchondral bone or articular cartilage. This result, consistent with the in vitro data, suggests that the elevation of COX2 levels in subchondral bone is a key mediator of HFD-induced subchondral bone alteration and osteoarthritis progression.

## Selective COX2 inhibitor use is associated with reduced risk of osteoarthritis progression and odds of BML structural damage worsening in humans with MetS-OA

To determine whether selective COX2 inhibitor also alleviates joint structural alterations in humans with MetS-OA, we conducted a longitudinal comparison of standard knee osteoarthritis outcomes between PS-matched COX2 inhibitor users and non-selective nonsteroidal anti-inflammatory drug (NSAID) users. Of 315 knees of COX2 inhibitor users and 488 knees of non-selective NSAID users, we selected 239 COX2 inhibitor users along with pair-matched 239 non-selective NSAID users using a 1:1 PS-matching method for potential confounders (*Supplementary file 1E*). The mean follow-up duration for standard knee OA outcomes was 4.4 years (median: 4 years, 1st and 3rd quartiles of 1 and 8 years, respectively). All analyses were further stratified according to the presence or absence of the MetS-OA phenotype (i.e., MetS-OA and no MetS-OA). *Supplementary file 1B* summarizes the baseline characteristics of included participants before and after applying PS matching. The mean (± standard deviation) age of the matched participants was 61 years±9 (61.3 years±8.8 in COX2 inhibitor users and 60.9 years±9.2 in non-selective NSAID users) and both groups were 73% female. The hazard of medial JSN OARSI osteoarthritis progression was significantly lower in COX2 inhibitor users compared with non-selective NSAID users, in participants with the MetS-OA phenotype (HR, 0.18; 95% CI, 0.04–0.85) but not in participants without

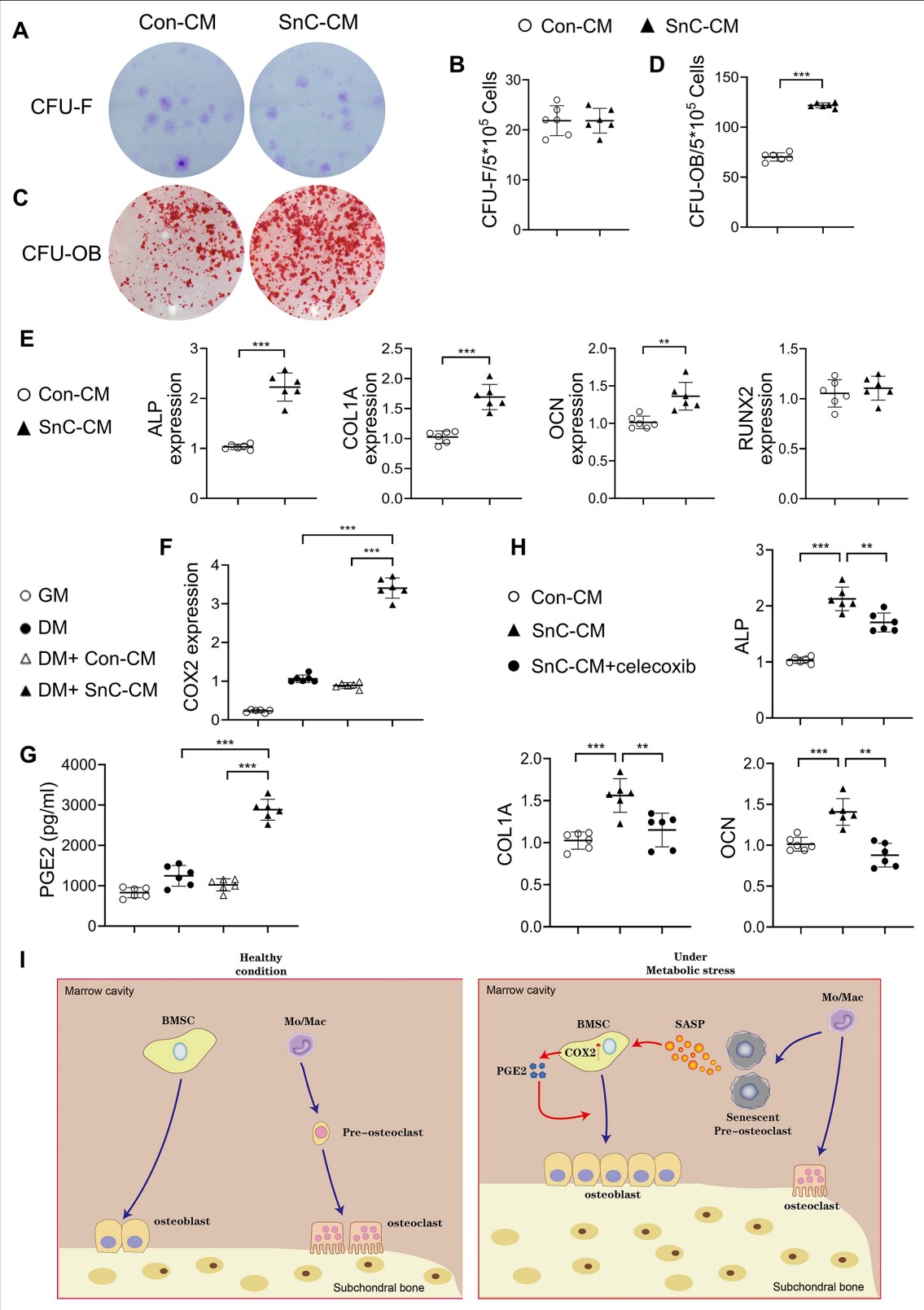

**Figure 7.** Secreted factors from preosteoclasts stimulate osteoblast differentiation through COX2-PGE2 signaling. (**A–D**) Representative images (**A, C**) and the quantified CFU-F frequency (**B**), and CFU-OB frequency (**D**) of bone marrow stromal cells treated with control conditioned medium (Con-CM) or senescent conditioned medium (SnC-CM). (**E**) qRT-PCR analysis of the relative levels of *Alpl, Col1a1, Bglap, Runx2* mRNA expression in bone marrow stromal cells cultured in the mixture of osteoblast differentiation medium (DM) and Con-CM or SnC-CM (DM:CM = 1:1). **p<0.01, and ***p<0.001.

*Figure 7 continued on next page*

*Figure 7 continued*

Statistical significance was determined by unpaired, two-tailed Student's *t*-test. (**F**) Relative levels of *Cox2* mRNA expression in BMSCs cultured in growth medium (GM), osteoblast DM, Con-CM, or SnC-CM. (**G**) PGE2 protein levels from GM, DM, Con-CM, or SnC-CM were calculated. Results are expressed as mean ± standard deviations. (**H**) qRT-PCR analysis of the relative levels of *Alpl, Col1a1, Bglap* mRNA expression in BMSCs cultured in the mixture of osteoblast DM and Con-CM or SnC-CM (DM:CM = 1:1), or together with celecoxib (40 μM). (**I**) Schematic model for the role of preosteoclast secretome-COX2/PGE2 axis in mediating subchondral bone formation during metabolic syndrome (MetS). Under MetS, preosteoclasts in subchondral bone marrow undergo cellular senescence and secrete SASP factors, which acts on both osteoclast precursors to suppress osteoclast differentiation and osteoblast precursors to activate COX2-PGE2 signaling to promote osteoblast differentiation for bone formation.

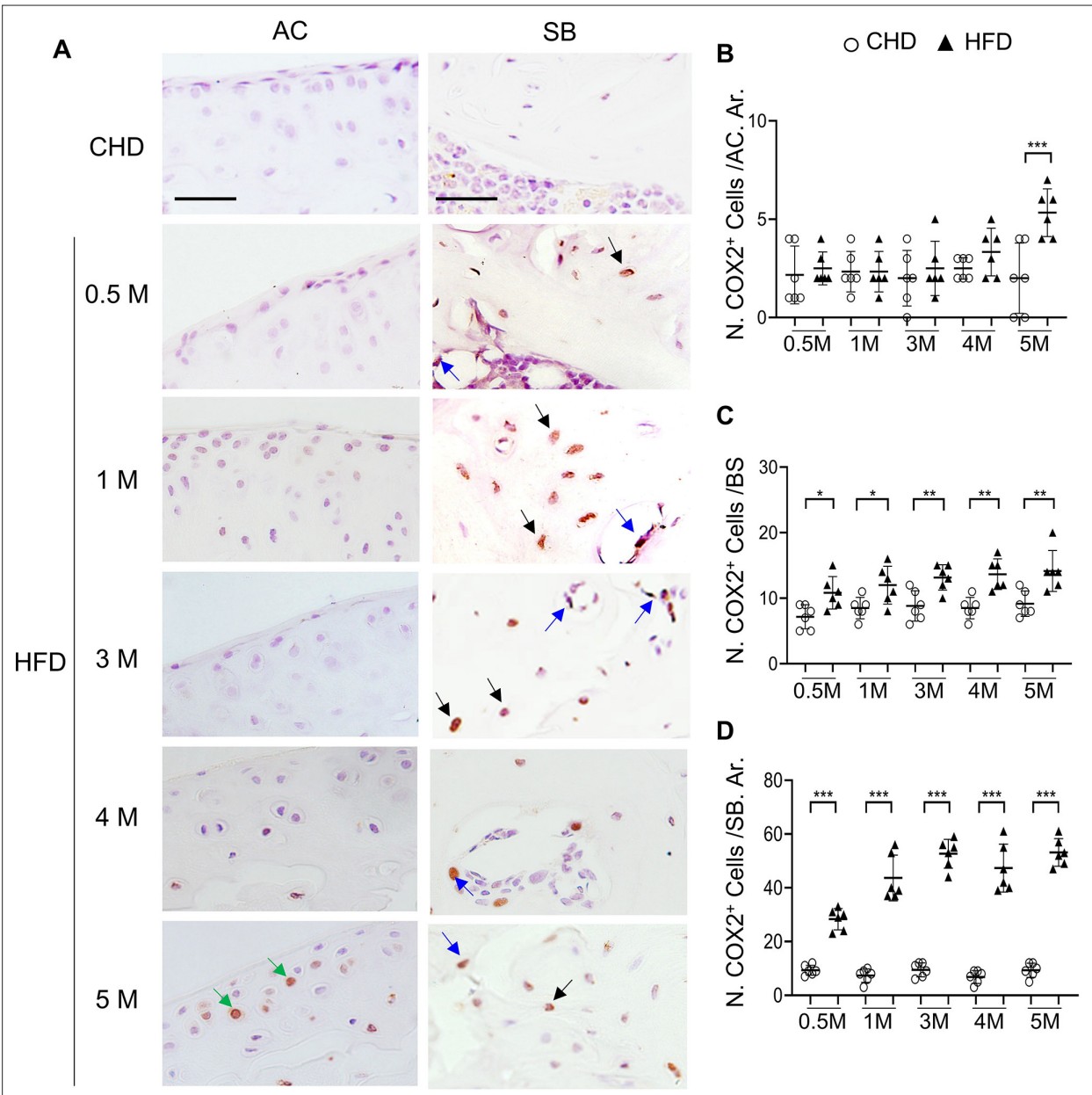

**Figure 8.** High-fat diet (HFD) mice have increased COX2+ cells in subchondral bone. Three-month-old C57BL/6 mice were fed a standard chow-food diet (CHD) or HFD for different time periods as indicated, n=6 mice per group. Immunofluorescence staining of knee joint tissue sections with antibody against COX2 (green) (**A**). Green arrows, COX2+ cells in cartilage; Blue arrows, COX2+ cells in bone surface osteoblasts; Black arrows, COX2+ cells in osteocytes. Quantification of the number of COX2+ cells per cartilage area (**B**), COX2+ cells per bone surface (**C**), and COX2+ cells per bone area (**D**). Results are expressed as mean ± standard deviation. n=5, *** p<0.001.

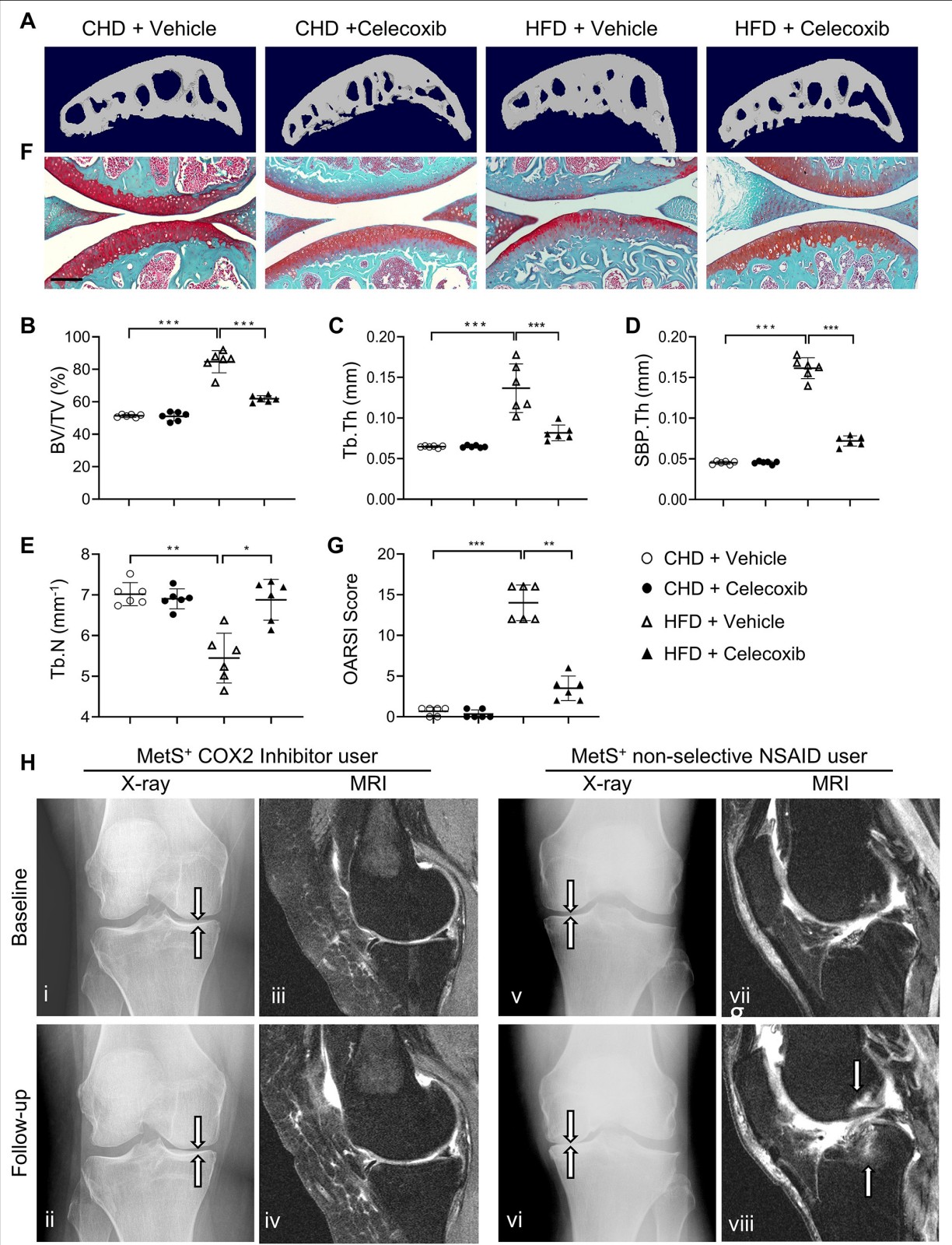

**Figure 9.** Cyclooxygenase 2 (COX2) inhibitor alleviates high-fat diet (HFD)-induced joint degeneration in mice. Three-month-old C57BL/6 mice were fed a standard chow-food diet (CHD) or HFD for 5 months. During the last 2 months of the HFD challenge, the mice also received celecoxib (16 mg/kg$^{-1}$ daily) or vehicle. n=6 mice per group. (**A–E**) Three-dimensional micro-computed tomography (μCT) images (**A**) and quantitative analysis of structural parameters of subchondral bone: bone volume/tissue volume (BV/TV, %) (**B**), trabecular thickness (Tb.Th, mm) (**C**), subchondral bone plate thickness

*Figure 9 continued*

(SBP. Th, mm) (**D**) and trabecular number (Tb.N, mm$^{-1}$) (**E**). (**F**) Safranin O-fast green staining of the tibia subchondral bone medial compartment (sagittal view). Scale bar, 200 μm (**G**) Calculation of Osteoarthritis Research Society International (OARSI) scores. *p<0.05, **p<0.01 and ***p<0.001. Statistical significance was determined by multifactorial ANOVA. All data are shown as means ± standard deviations. (**H**) COX2 inhibitor user (**i–iv**) and nonselective nonsteroidal anti-inflammatory drug (NSAID) user (v-viii) baseline and follow-up radiographs and magnetic resonance images of patients with metabolic syndrome (MetS)-associated osteoarthritis phenotype (MetS$^+$). Weightbearing posteroanterior radiograph of the right knee using fixed flexion protocol of a 64-year-old MetS$^+$ woman with COX2 inhibitor use at baseline (**i**) and at 24-month follow-up (ii). Sagittal intermediate-weighted fat-suppressed MRI sequences in the same knee of COX2 inhibitor user patient at baseline (iii) and at 24-month follow-up (iv). Weightbearing posteroanterior radiograph of the left knee using fixed flexion protocol of a 62-year-old MetS$^+$woman with nonselective NSAID use at baseline (**v**) and at 24-month follow-up (vi). Sagittal intermediate-weighted fat-suppressed MRI sequences in the same knee of nonselective NSAID use patient at baseline (vii) and at 24-month follow-up (viii).

MetS-OA (HR, 0.96; 95% CI, 0.53–1.74) (*Table 2*). No significant difference was found between COX2-inhibitor users and non-selective NSAID users in knee osteoarthritis incidence or symptoms of knee osteoarthritis in either study strata. (*Table 2*). We further compared 24 month worsening in subchondral BML structural damage between COX2 inhibitor users and non-selective NSAID users in participants with the MetS-OA phenotype. Our results showed that MetS-OA COX2 Inhibitor users, compared to non-selective NSIAD users, had lower odds of subchondral damage worsening. This was evident as lower odds of worsening in the number of affected knee subregions with BMLs (OR, 0.35; 95% CI, 0.13–0.93) and lower odds of worsening in the BML scores (OR, 0.45; 95% CI, 0.20–0.99). (*Table 3*).

A representative knee radiograph showed grade 2 medial JSN (using the OARSI grading method) with 3.5 mm of joint space width in a 64-year-old woman with MetS-OA phenotype (MetS$^+$) at baseline (*Figure 9Hi*). After 24 months of COX2 inhibitor use, follow-up radiographs of the same knee showed minimal progression of medial JSN, as evidenced by no interval change in OARSI grading and minimal narrowing of the joint space width to 2.9 mm (*Figure 9Hii*). Conversely, a knee radiograph of a 62-year-old MetS$^+$ woman showed grade 2 JSN with minimal (3.7 mm) joint space width at baseline (*Figure 9Hv*); however, a follow-up radiograph of the same knee showed progression of JSN, as evidenced by interval change in OARSI grading to grade 3 with a joint space width of 1.0 mm (*Figure 9Hvi*). Consistently, magnetic resonance imaging (MRI) showed a significant increase in the size and number of subchondral bone marrow lesions in the knee of a non-selective NSAID user (*Figure 9Hviii vs. vii*) but unchanged subchondral bone marrow lesion size and number in the knee of a selective COX2 inhibitor user (*Figure 9Hiv vs. iii*). The decreased risk of medial JSN OARSI osteoarthritis progression in COX2 inhibitor users with MetS-OA phenotype is consistent with our findings in animal models of MetS-OA.

**Table 2.** Longitudinal comparison of standard knee OA outcomes between human COX2 inhibitor users vs. non-selective NSAID users according to the presence of MetS.

| | COX2I users vs. non-selective NSAID users Hazard ratio (95% Confidence Interval), p-value, Sample size, Number of events[*] [COX2I: NSAID] | |
| --- | --- | --- |
| | MetS-OA$^+$ | MetS-OA$^-$ |
| Knee OA incidence | 0.42 (0.04–4.77), p:0.487, N: 30 (15:15), Events[*] [2:3] | 0.62 (0.13–2.84), p:0.537, N: 118 (59:59), Events[*] [5:9] |
| Knee OA progression | 0.18 (0.04–0.85), p:0.030, N: 94 (47:47), Events[*] [2:8] | 0.96 (0.53–1.74), p:0.886, N: 266 (133:133), Events[*] [21:23] |
| Symptomatic incidence (NASS) | 0.8 (0.27–2.4), p:0.689, N: 108 (54:54), Events[*] [2:3] | 1.56 (0.63–3.89), p:0.337, N: 324 (162:162), Events[*] [18:12] |

COX2I: cyclooxygenase 2 inhibitor, MetS: metabolic syndrome, NASS: non-acceptable symptomatic state, NSAID: Non-Steroidal Anti-Inflammatory Drug, OA: Osteoarthritis. Standard OA outcomes were compared between knees of COX2 inhibitor users vs. matched non-selective NSAID users. Analysis was stratified analysis for the presence of MetS-associated OA (MetS-OA$^+$ vs. MetS-OA$^-$). Cox proportional hazards were used. Knees of participants were matched for confounders using the 1:1 propensity-score matching method. Events are knee OA incidence defined by Kellgren-Lawrence (KL) grade ≥2 in participants with KL equal to 0–1, knee OA progression defined by partial or whole grade progression in Osteoarthritis Research Society International medial joint space narrowing grade, and knee OA symptomatic incidence measured by NASS. The mean follow-up duration for standard knee OA outcomes was 4.4 years (median: 4 years, 1st and 3rd quartiles of 1 and 8 years). N corresponds to the total number of knees included in each analysis and the number of matched knees of COX2 inhibitor users vs. non-selective NSAID users in the parentheses.

[*] Number of events for each outcome has been shown separately in the brackets for COX2 inhibitor users and non-selective NSAID users.

**Table 3.** Longitudinal comparison of subchondral BML worsening between human COX2 inhibitor users vs. non-selective NSAID users with MetS-OA.

| Subchondral BML Worsening (MOAKS) | COX2 inhibitor users vs. non-selective NSAID users with MetS-OA | |
|---|---|---|
| | Odds ratio (95% Confidence Interval), p-value, Sample size | Number of events in each group [COX2 inhibitor: NSAID]* |
| Worsening in number of affected subregions with BML | 0.35 (0.13–0.93), p:0.035, N: 88 (44:44) | Improvement, [3:2] No change, [14:9] Worsening, [27:33] |
| Improvement in number of affected subregions | 1.21 (0.19–7.72), p:0.839, N: 88 (44:44) | Yes, [3:2] |
| Maximum worsening in BML score | 0.45 (0.2–0.99), p:0.046, N: 88 (44:44) | No change, [18:9] Worsening by ≤1 grade, [22:27] by ≥2 grades, 4:8 |
| Improvement in the subregions' BML score | 1.28 (0.33–5.00), p:0.722, N: 88 (44:44) | Yes, [9:6] |

Among the PS-matched COX2 inhibitor users and non-selective NSAID users with MetS-OA, participants with available baseline and 24-month follow-up MRIs were included. A musculoskeletal radiologist read and scored MRIs according to validated MOAKS measures of subchondral BML worsening. Logistic mixed-effect regression models were used for subchondral BML assessments. All analyses were adjusted for the baseline Kellgren-Lawrence (KL) and Osteoarthritis Research Society International medial joint space narrowing (OARSI JSN) grades of knees. Subchondral BML worsening was assessed using standard MOAKS measures. N corresponds to the total number of knees included in each analysis and the number of matched knees of MetS-OA and PTOA participants in the parenthesis.

* Number of events for each outcome has been shown separately in the brackets for COX2 inhibitor users and non-selective NSAID users.

BML: Bone marrow lesion, MetS: metabolic syndrome, MOAKS: MRI Osteoarthritis Knee Score, NSAID: Non-Steroidal Anti-Inflammatory Drug, OA: osteoarthritis.

## Discussion

In clinical practice, osteoarthritis has various causes (*Misra et al., 2015*) (e.g. PTOA, non-traumatic age-associated, and MetS-OA) and presents at different stages (early vs. late). In animal studies, increasing evidence supports an important role of subchondral bone changes during PTOA progression. However, almost nothing is known about progressive changes in subchondral bone in nontraumatic osteoarthritis. Our findings show that humans and mice with MetS-OA have a subchondral bone phenotype distinct from that of PTOA and have a greater likelihood of developing osteoarthritis-related subchondral bone damage. Structurally, contrary to the increased osteoclast bone resorption in subchondral bone in early-stage PTOA (*Zhen et al., 2013*; *Sun et al., 2021*; *Zhu et al., 2019*), there is rapid thickening of subchondral bone plate and trabecular bone in HFD-challenged mice and STR/Ort mice, both of which are MetS-OA mouse models. These subchondral alterations appear much earlier than the occurrence of cartilage degradation. Cellularly, unlike the accumulated SnCs in cartilage and synovium in PTOA (*Jeon et al., 2017*; *Jeon et al., 2018*), increased SnCs are located almost exclusively in the subchondral bone in MetS-OA mice. We also found that many of the SnCs were RANK⁺TRAP⁺ preosteoclasts in bone marrow. Mechanistically, the senescent preosteoclasts acquire a unique secretome, which acts on both osteoclast and osteoblast lineages in a paracrine manner, leading to inhibited osteoclast bone resorption and increased osteoblastic bone formation for rapid subchondral bone thickening at the onset of osteoarthritis.

Accumulated senescent chondrocytes in articular cartilage have been found in PTOA (*Jeon et al., 2017*; *Diekman, 2018*). We observed significantly increased senescent preosteoclasts in subchondral bone but not in articular cartilage in HFD or STR/Ort mice. Therefore, the joint locations of the SnCs and the major SnC type are distinct in MetS-OA and PTOA. We also show that deletion of senescence gene *Cdkn2a* specifically in osteoclast lineage attenuated short term HFD challenge-induced pathological subchondral bone alterations. As a result, HFD-induced cartilage degeneration was also greatly alleviated. Therefore, preosteoclast senescence in subchondral bone is a key mediator for the progression of MetS-OA. Our data showing that accumulation of senescent preosteoclasts and subchondral bone architectural change occur rapidly (0.5 months) after HFD treatment but cartilage degeneration occurred much later (5 months) after HFD treatment (*Figures 1 and 3*) suggests that the eventual cartilage damage may be caused by progressive subchondral bone architectural changes rather than a direct effect from the SASP of preosteoclasts. The contribution of aberrant subchondral bone alteration to articular cartilage degeneration has been well-recognized. Accumulating evidence suggests that normal subchondral bone structure is essential for the homeostasis of articular cartilage

(*Zhen et al., 2021*; *Goldring and Goldring, 2016*), and the incremental increase of either subchondral bone/plate thickness or subchondral plate stiffness modulus subsequently increases stress and creates uneven stress distributions in the overlaying articular cartilage for its degradation (*Zhen et al., 2021*; *DeFrate et al., 2019*; *Lories and Luyten, 2011*). Moreover, strategies reducing/normalizing subchondral bone alterations during the early stage of PTOA effectively prevented cartilage degeneration (*Su et al., 2020*; *Zhen et al., 2013*). It is of interest to define whether increases in subchondral bone volume and subchondral plate/trabecular thickness result in incremental increase in mechanical stress on cartilage for its degeneration in MetS-OA.

Our in vivo results from HFD-challenged mice and in vitro evidence from cultured preosteoclasts showed that senescent preosteoclasts secrete previously identified common SASP factors, such as IL-1β, IL-6, VEGF, and OPN (*Coppé et al., 2010*; *Tchkonia et al., 2013*; *Shang et al., 2020*; *Gómez-Santos et al., 2020*), as well as several factors that were not previously recognized in SnCs, including Lipocalin-2, Resistin, Cystatin C, IL-33, CCN4, MPO, and PDGF-BB. Importantly, the abnormally high production of these factors in the subchondral bone of HFD-challenged mice is fully or partially rectified by deletion of *Cdkn2a* from preosteoclasts. It is well-accepted that the SASP is largely distinct in composition, highly cell type–specific, and also dynamic depending on the senescence inducers (*Basisty et al., 2020*). The SASP secretome promotes, via autocrine/paracrine pathways, the reprogramming of neighboring cells and modifies the microenvironment (*Mosteiro et al., 2016*). Here, we show that the secretome of preosteoclasts negatively regulates the differentiation of osteoclasts but promotes osteoblast differentiation in the metabolic dysregulation–associated microenvironment of subchondral bone/bone marrow. These effects of the SASP factors appear opposite of what have been found in bone in the context of aging. Particularly, it has been reported that the SASP in bone microenvironment impair osteoblastic bone formation and enhance osteoclastic bone resorption during aging (*Farr et al., 2017*; *Farr et al., 2016*). The contradicting results is likely attributed to the different composition and activity of subchondral preosteoclast SASP in the setting of MetS-OA than in the setting of aging. Of note, among the preosteoclast-derived factors that we identified, there are both osteoclast differentiation stimulators, such as IL-1β (*Ruscitti et al., 2015*) and IL-6 (*Rose-John, 2018*), and inhibitors, such as MPO (*Zhao et al., 2021*), IL-33 (*Amarasekara et al., 2018*), and CCN4 (*Chang et al., 2018*). The net result of these 2 opposite effects is likely an inhibition of osteoclastic bone resorption given our in vivo observation of the reduction in osteoclast numbers in subchondral bone of HFD mice and in vitro finding of the partial inhibition of osteoclastogenesis by the SnC-CM. Therefore, the SASP factors from senescent preosteoclasts have a unique paracrine effect on the nearby non-senescent osteoclast precursors to inhibit their osteoclastogenic ability. Moreover, our RNA-seq data also showed many downregulated osteoclast differentiation genes in the senescent preosteoclasts relative to non-senescent cells, suggesting that the senescent preosteoclasts have decreased ability to further differentiate into mature osteoclasts. An unexpected effect of the preosteoclast secretome is to stimulate osteoblast differentiation of the BMSCs. Of note, the CFU-F and gene expression of RUNX2, a transcription factor controlling the early commitment of BMSCs to osteoblast lineage, were unchanged in cells treated with SnC-CM relative to those treated with Con-CM. Thus, the SASP secretome specifically promotes osteoblast differentiation/maturation without affecting the colony forming capacity and lineage commitment of BMSCs. Importantly, our in vivo data demonstrate that unlike bone surface localization in normal physiological conditions, osteoblasts were aberrantly accumulated and formed clusters in the marrow cavity of HFD mice. This in vivo finding is consistent with the increased osteoblast differentiation of the BMSCs in response to SnC-CM. The formation of osteoblast clusters within the bone marrow may represent a unique subchondral bone feature of MetS-OA, contributing to the rapid development of subchondral plate thickening and sclerosis.

Our work shows that COX2-PGE2 activation is a key mediator of the SASP secretome–induced osteoblast differentiation and subchondral bone thickening that occurs in MetS. Of note, all of the identified SASP factors produced by senescent preosteoclasts, including IL-1β, IL-6, OPN (*Jain et al., 2006*), IL-33 (*Li et al., 2018*), Lipocalin-2 (*Hamzic et al., 2013*), MPO (*Panagopoulos et al., 2017*), VEGF (*Chien et al., 2009*), Resistin (*Su et al., 2017*), and PDGF-BB (*Englesbe et al., 2004*), were identified previously as positive COX2-PGE2 stimulators. Indeed, both COX2 gene expression in BMSCs and PGE2 level in the culture medium were upregulated when the cells were incubated with SnC-CM relative to Con-CM. Addition of selective COX2 inhibitor celecoxib in the BMSC culture medium

significantly downregulated the osteoblast differentiation marker genes stimulated by SnC-CM, confirming the requirement of COX2-PGE2 activation for SASP-induced osteoblast differentiation of the BMSCs. Further, our in vivo data also show that the majority of COX2[+] cells were subchondral osteoblasts and osteocytes during the progression of MetS-OA, and selective COX2 celecoxib treatment markedly alleviated pathological subchondral bone thickening and osteoarthritis progression. Thus, COX2-PGE2 is a critical mediator of subchondral bone alteration and disease progression of MetS-OA. Clinically, COX2 inhibitors have been used widely to relieve arthritis-associated pain at lower doses and to inhibit inflammation at higher doses (*Geba et al., 2002*). However, there are contradictory reports regarding their direct disease-modifying function (*Zweers et al., 2011*). It is unclear from multiple clinical trials and preclinical studies during the past two decades whether selective COX2 inhibitors protect cartilage and slow osteoarthritis progression. One of the main reasons for the inconclusive results of these studies could be the heterogeneous causes of osteoarthritis. Our findings from the human OAI dataset analysis confirmed the central role of subchondral bone damage in MetS-OA worsening. We showed that participants with MetS-OA with similar longitudinal worsening to their matched PTOA participants had increased subchondral bone damage. Intriguingly, among selective COX2-inhibitor users, only those with MetS-OA have alleviated joint structural alterations, and use of COX2-inhibitor is associated with reduced the odds subchondral bone damage. More importantly, this effect is seen compared with participants who use other analgesics, even after matching for pain status (which can be a source for confounding by indication bias, as the most widely indication of both non-selective NSAID and selective COX2 inhibitor prescription). These findings suggest that selective COX2 inhibitors but not the non-selective NSAIDs may have disease-modifying properties for MetS-OA.

## Acknowledgements

The authors thank Yasuhiro Kobayashi (Matsumoto Dental University, Japan) and Gloria H Su (Columbia University Medical Center) for kindly providing the *Tnfrsf11a*[Cre/+] (RANK-Cre) mice and the *Cdkn2a*[flox/flox] (p16[flox/flox]) mice, respectively. We acknowledge the assistance of The Johns Hopkins School of Medicine Microscope Facility. The authors also acknowledge the assistance of Rachel Box, Jenni Weems, and Kerry Kennedy at The Johns Hopkins Department of Orthopaedic Surgery Editorial Services for editing the manuscript. This work was supported by the National Institutes of Health grant R01AG068226 and R01AG072090 to MW, R01AR079620 to SD, and P01AG066603 to XC.

## Additional information

### Competing interests

Ali Guermazi: received consultancy fees from Pfizer, Novartis, MerckSerono, TissueGene, AstraZeneca, and Regeneron. The author has no other competing interests to declare. Mei Wan: Reviewing editor, eLife. The other authors declare that no competing interests exist.

### Funding

| Funder | Grant reference number | Author |
| --- | --- | --- |
| National Institutes of Health | R01AG068226 | Mei Wan |
| National Institutes of Health | R01AG072090 | Mei Wan |
| National Institutes of Health | R01AR079620 | Shadpour Demehri |
| National Institutes of Health | P01AG066603 | Xu Cao |

The funders had no role in study design, data collection and interpretation, or the decision to submit the work for publication.

## Author contributions
Weiping Su, Guanqiao Liu, Data curation, Software, Formal analysis, Investigation, Methodology; Bahram Mohajer, Data curation, Software, Formal analysis, Validation, Investigation, Methodology, Writing – original draft; Jiekang Wang, Data curation, Software, Methodology; Alena Shen, Data curation, Writing – review and editing; Weixin Zhang, Data curation, Methodology; Bin Liu, Software, Formal analysis; Ali Guermazi, Peisong Gao, Conceptualization, Writing – review and editing; Xu Cao, Conceptualization, Funding acquisition, Writing – review and editing; Shadpour Demehri, Conceptualization, Resources, Supervision, Funding acquisition, Investigation, Writing – original draft, Project administration, Writing – review and editing; Mei Wan, Conceptualization, Resources, Supervision, Funding acquisition, Validation, Visualization, Writing – original draft, Project administration, Writing – review and editing

## Author ORCIDs
Xu Cao ⓘ http://orcid.org/0000-0001-8614-6059
Shadpour Demehri ⓘ http://orcid.org/0000-0001-5991-5924
Mei Wan ⓘ http://orcid.org/0000-0001-9404-540X

## Ethics
Human subjects: We used data from the longitudinal multi-center OAI study (2004-2015 clinicaltrials. gov identifier: NCT00080171). All 4,796 enrolled patients gave written informed consent. Institutional review boards of four OAI collaborating centers have approved the OAI study's Health Insurance Portability and Accountability Act-compliant protocol (approval number: FWA00000068).

## Decision letter and Author response
Author response https://doi.org/10.7554/eLife.79773.sa2

---

# Additional files

## Supplementary files
• Supplementary file 1. Human osteoarthritis initiative datasets used in the study. (A) Baseline characteristics of the participants according to presence of metabolic syndrome-associated OA (MetS⁺ PTOA⁻ versus PTOA⁺ MetS⁻) before and after propensity score matching.(B) Baseline characteristics of human COX2 inhibitor and non-selective NSAID users included in the study, before and after propensity score matching. Matched participants were includde in the analysis of COX2 inhibitor use association with OA outcomes, according to its phenotype.(C) Osteoarthritis Initiative (OAI) datasets used in the study. (D) Flowchart outlining the selection criteria and PS-matching process according to the presence of metabolic syndrome-associated OA (MetS-OA) and post-traumatic OA (PTOA) in Osteoarthritis initiative participants. (E) Flowchart outlining the selection criteria and PS-matching process of human COX2 inhibitor and non-selective NSAID users from the Osteoarthritis initiative dataset.

• MDAR checklist

## Data availability
The data that support the findings of this study are available within the article and Supplementary file. Sequencing data have been deposited in Dryad and can be acquired through online portal at https://doi.org/10.5061/dryad.q2bvq83n6. The naming and version of OAI dataset files used in our study are listed in Supplementary file 1C and can be acquired through OAI online portal at https://nda.nih.gov/oai.

The following dataset was generated:

| Author(s) | Year | Dataset title | Dataset URL | Database and Identifier |
|---|---|---|---|---|
| Su W | 2022 | Senescent preosteoclast secretome promotes metabolic syndrome-associated osteoarthritis through COX2-PGE2 | https://doi.org/10.5061/dryad.q2bvq83n6 | Dryad Digital Repository, 10.5061/dryad.q2bvq83n6 |

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
