## [Editor Report]

The manuscript presents novel findings that link, in mechanistic terms, metabolic syndrome with osteoarthritis. A central role of a senescent preosteoclast secretome-COX2/PGE2 axis has been established. The translational significance relates to the future use of selective COX2 inhibitors as disease-modifying agents in the osteoarthritis that accompanies metabolic syndrome.

---

## [Author Response]

[Editors' note: we include below the reviews that the authors received from another journal, along with the authors’ responses.]

Point-by-point response to reviewers’ commentsWe would like to thank the reviewers for their thoughtful and constructive comments regarding our manuscript. We have addressed all of the questions and concerns brought forth through additional experimentation and clarification. Specifically, in the mouse study, the reviewers (Reviewer A and B) mainly concerned about the changes (i.e. cellular senescence) of articular cartilage during metabolic syndrome associated osteoarthritis (MetS-OA) and how the subchondral preosteoclast senescence and bone alterations lead to cartilage degeneration. We have conducted more immunostaining analyses to demonstrate the change in senescent cells in both subchondral bone and articular cartilage during the progression of MetS-OA. More importantly, we have demonstrated that senescent preosteoclasts contribute to the progression of cartilage degeneration using a new genetic mouse model to delete the key senescence gene specifically in osteoclast lineage. In the human study, the major concern raised by the reviewers (Reviewer B and C) is that the results of the human participants with MetS-OA are insufficient to prove the role of subchondral bone alterations in OA progression as shown in the MetS-OA mouse models. To further similarize our human OA participants with corresponding mouse models of OA, using propensity score matching, we have further compared subchondral bone marrow lesions worsening between MetS-OA participants (without PTOA) to PTOA participants (without MetS-OA) akin to the mice models to better delineate the unique role of subchondral bone damage in MetS-OA.To aid in readability, we have made the modified text in blue font to distinguish from the black original text. The following responses have been prepared to address all of the reviewers’ comments in a point-by-point fashion.Response to comments from Reviewer A:In this study, Su et al. examine the pathogenesis of metabolic syndrome-associated OA (MetS-OA) using a combination of human and mouse/in vitro data. Their findings implicate senescent pre-osteoclastic cells and their SASP in inhibiting osteoclasts and stimulating osteoblasts in the subchondral bone, leading to changes of OA.Overall, the studies are done well and the results support the conclusions. To the authors’ credit, they use multiple approaches to establish the presence of senescent pre-osteoclasts (increased p16 expression, SA-βGal staining, redistribution of HMGB1 from the nucleus to the cytoplasm, loss of lamin B1). I have the following points for the authors to address:1. The authors focus exclusively on subchondral bone changes in their mouse models. It would be important to define what is happening in the cartilage during the time that the bone changes are occurring in their models. Are cartilage cells becoming senescent? Is there evidence for damage to the articular cartilage from the SASP? Although OA is associated with sub-chondral bone changes, the ultimate problem is the deterioration of the articular cartilage, and that seems to be ignored here.

We recognize the importance of this question and have conducted a panel of new experiments to define the cartilage changes when the alterations of subchondral bone occurred in our model system. We have first evaluated the senescent cells on the joint cartilage tissue in our HFD-induced osteoarthritis (OA) mouse model. Fluorescence imaging of the knee joint tissue sections from *p16^tdTom^* reporter mice shows that tdTom+ cells were exclusively localized at subchondral bone/bone marrow at earlier time points (1 and 3 months) after the mice were fed HFD (Figure 3, A and C). tdTom+ SnCs were not seen in articular cartilage until later (5 months) after HFD challenge (Figure 3, A and B), when significant cartilage degeneration occurs as indicated by a markedly increased OARSI score (Figure 1, A and B). Consistently, increased SA-βGal^+^ cell number were detected at the subchondral bone/bone marrow (Figure 3, D and F) but not in articular cartilage (Figure 3, D and E) in mice fed a HFD for 1 month or 3 months. Our results suggest that senescent cells primarily accumulate in subchondral bone at the pre- or early-OA stage and appear in cartilage only at more advanced stages.

We then examined whether cellular senescence in subchondral bone may be a major pathogenic culprit of the eventual cartilage degeneration induced by metabolic syndrome by characterizing joint phenotype of a conditional *RANK-Cre; p16 ^flox/flox^* (p16^cKO^) mouse line, in which senescence gene *p16^INK4a^* is deleted in RANK^+^ cells. *RANK-Cre* (kindly provided by Dr. Y. Kobayashi at Matsumoto Dental University, Japan) targets osteoclast lineage cells (*Nat Med.* 2012; 18: 405–412; *Nat Med.* 2016; 22: 1203–1205). Consistent with Figure 3, D-F, SA-βGal+ cells accumulated only in subchondral bone marrow in WT mice with 1-month HFD treatment, and this increase in subchondral SA-βGal+ cells was greatly dampened in the p16^cKO^ mice relative to WT mice (Figure 5—figure supplement 7). Therefore, deletion of *p16^INK4a^* in RANK^+^ cells efficiently prevents/blocks subchondral cellular senescence. Importantly, deletion of senescent preosteoclasts in subchondral bone indeed prevents the increase of subchondral thickness (Figure 5, A-H) and greatly attenuated cartilage degeneration (Figure 5, I-L) induced by HFD treatment. Collectively, the results from the new panel of experiments suggest that preosteoclast senescence in subchondral bone is a key initiating factor for the pathological subchondral bone architectural alterations, which may eventually lead to cartilage degeneration.

Whether bone preosteoclast SASP directly causes pathological chondrocyte changes for articular cartilage degeneration is an interesting, unresolved question. Our data showing that accumulation of senescent preosteoclasts and subchondral bone architectural change occur rapidly (0.5 months) after HFD treatment but cartilage degeneration occurred much later (5 months) after HFD treatment (Figure 1 and 3) suggests that the eventual cartilage damage may be caused by progressive subchondral bone architectural changes rather than a direct effect from the SASP of preosteoclasts. The contribution of aberrant subchondral bone alteration to articular cartilage degeneration has been well-recognized in the OA field. Accumulating evidence suggest that normal subchondral bone structure is essential for the homeostasis of articular cartilage (*Nat Commun.* 2021. 12: 1706; *Nat Rev Rheumatol*. 2016. 12: 632-644), and the incremental increase of either subchondral bone/plate thickness or subchondral plate stiffness modulus subsequently increases stress and creates uneven stress distributions in the overlaying articular cartilage for its degradation (*Osteoarthritis Cartilage* 2019. 27: 392-400; *Nat Commun*. 2021. 12: 1706). Moreover, strategies reducing/normalizing subchondral bone alterations during the early stage of post-traumatic OA (PTOA) effectively prevented cartilage degeneration (*JCI Insight.* 2020. 5:e135446; *Nat Med* 2013;19:704-712). It is of interest in the future to define whether increases in subchondral bone volume and subchondral plate/trabecular thickness result in incremental increase in mechanical stress on cartilage for its degeneration in MetS-OA. The discussion on this point has also been added in the revised manuscript (Line 425-442).

2. Their demonstration that the SASP of senescent pre-osteoclastic cells inhibits osteoclasts and promotes osteoblasts is the opposite of what other studies have found, specifically in the context of aging, where the SASP from senescent cells stimulates osteoclasts and inhibits bone formation. The most obvious explanation is that the SASP in the setting of MetS-OA is different than that in the setting of aging. But this point should be explicitly addressed in the Discussion, as otherwise this discrepancy may lead to confusion in the literature.

We appreciate the valuable comment and agree with the reviewer that it is important to clarify the discrepancy of our results and previous findings regarding the SASP effects on osteoclast and osteoblast lineage cells. Our results show that the senescent preosteoclast secretome negatively regulates the differentiation of osteoclasts but promotes osteoblast differentiation in the metabolic dysregulation–associated microenvironment of subchondral bone/bone marrow. These effects of the SASP factors appear opposite of what have been found in bone in the context of aging. Particularly, it has been reported that the SASP in bone microenvironment impair osteoblastic bone formation and enhance osteoclastic bone resorption during aging (*Nat Med* 23, 1072-1079; *J Bone Miner Res* 31, 1920-1929). We fully agree with the reviewer that the contradicting results is likely attributed to the different composition and activity of subchondral preosteoclast SASP in the setting of MetS-OA than in the setting of aging. Indeed, we have identified several new SASP factors, including Lipocalin-2, Resistin, Cystatin C, IL-33, CCN4, MPO, and PDGF-BB (Figure 6), that were not previously recognized in SnCs in bone in the setting of aging. Some of these factors, such as IL-33 and CCN4, have potent inhibitory effect on osteoclast differentiation (*J Immunol* 2011;186:6097-6105; *J Biol Chem* 2015;290:14004-14018). All of them are positive activators of COX2-PGE2 signaling, which acts on osteoblastic precursors to stimulate osteoblast differentiation (*J Cell Biochem* 2006;99:824-834; *Biochem Biophys Res Commun* 2007;360:199-204; Immune Netw 2018;18:e8; *Int J Oncol.* 2017;50(4): 1191-1200; *J Clin Invest* 2019;129:2578-2594; *J Neuroendocrinol.* 2013;25: 271-280; *J Bone Miner Res* 2011;26:193-208). Importantly, the results from our in vitro cell culture (Figure 7 and Figure 6—figure supplement 8) and in vivo mouse studies (Figure 5) consistently demonstrate that the SASP of senescent preosteoclasts inhibits osteoclast differentiation and promotes osteoblast differentiation and bone formation in subchondral bone in MetS-OA. Moreover, the new RNA-seq data revealed many downregulated osteoclast differentiation genes in the senescent preosteoclasts relative to non-senescent cells (Figure 6—figure supplement 10 and 11), suggesting that the senescent preosteoclasts also have decreased ability to further differentiate into mature osteoclasts. The discussion on this point has been added in the revised manuscript (Line 449-481).

Aging in mice contributes to the development of spontaneous OA, similar to what occurs in humans. For example, C57BL/6 mice usually develop knee OA in the absence of surgical knee operation at about 17 months of age. Significant ongoing effort in the lab is to determine the subchondral bone phenotypic changes in the setting of aging. We found that unlike increased subchondral bone volume and subchondral plate thickness in MetS-associated OA, there is a decrease in subchondral bone volume and trabecular thickness in aged mice relative to young mice (see Author response image 1), indicating increased subchondral osteoclast bone resorption and/or reduced osteoblast bone formation in age-associated OA joints. We will further characterize the senescent cells and the SASP in subchondral bone/bone marrow in the setting of aging. These experiments will be of great interest in understanding the distinct role of SASP in subchondral bone changes in different OA subtypes and will be an important area of future focus in the laboratory.

**Author response image 1. sa2fig1:** Subchondral bone changes in aged mice relative to young mice.

3. A brief introduction to the STR/Ort mice should be provided in the Introduction for readers who may not be familiar with this model.

As suggested, an introduction of the STR/Ort mice have been included in the Introduction section (Line 78-82).

4. In each of the studies, the sex of the mice should be indicated. Were the mice in the different groups matched for sex, as MetS-OA may be different in male vs female mice (or humans)?

In the animal studies, we used male mice in most of the experiments as we were characterizing a new subtype of OA mouse models and wished to first address the effects of MetS while excluding biological sex as a confounding variable. Now, however, we have included additional data in the subchondral bone assessments using female HFD mice. As noted in the updated figures, female mice fed HFD for 3 months had the same subchondral bone phenotype (i.e. dramatically increased subchondral bone volume, subchondral plate thickness, and trabecular thickness) (Figure 1—figure supplement 3) as did in the male mice fed HFD relative to the chow diet mice (Figure 1). The results suggest that HFD challenge induces similar alterations in subchondral bone in male and female mice. As for the STR/Ort mice, it has been recognized that male mice have a higher incidence of OA than female mice (*Osteoarthritis Cartilage* 2017. 25: 802-808). We therefore used male STR/Ort and CBA control mice in this study. We look forward to future studies examining in more detail on the comparison of the female and male mice regarding the role of cellular senescence in the development of MetS-OA. Indeed, we are currently testing whether subchondral preosteoclasts become senescent and have the same SASP in female MetS mice as those from the male MetS mice.

In the human participants assessments, both female and male human subjects were included, and the sex of the participants is propensity-score matched between comparison groups and are presented in Supplementary Tables 1 and 2 (information on sex of the participants is in blue font in these 2 tables in the manuscript). In addition, we also added a description of participants' sex in the manuscript text Results section (Line 99-102 and line 376).

Response to comments from Reviewer B:In this paper the authors investigate the pathogenic mechanisms of metabolic syndrome-associated osteoarthritis (MetS-OA), which is a distinct type of OA. By using two different mouse models of metabolic syndrome/OA (high fat diet and STR/Ort mice), they found that a rapid increase in joint subchondral bone plate and trabecular thickness occurs before cartilage degeneration. This increase in bone mass is attributed to increased osteoblast number in the subchondral bone. They show that the increase in osteoblasts is due to senescent pre-osteoclasts that secrete senescence- associate secretory phenotype (SASP) factors which activate Cyclooxygenase 2 transcription and prostaglandin 2 (PGE2) production in osteoblasts. PGE2 stimulates the differentiation of osteoblasts progenitors and bone formation in the subchondral bone. Celecoxib, a COX 2 inhibitor, attenuates subchondral bone formation and progression of OA associated with metabolic syndrome in both mice models.The studies in humans indicate that patients with metabolic syndrome have increased progression of OA during longitudinal follow up, compared to matched patients without metabolic syndrome; more importantly, the use of COX2 inhibitors reduced OA progression in patients with metabolic syndrome but not in matched patient without metabolic syndrome. MRI imaging, shown in just 1 patient/group, shows the presence of significant increase in size of the subchondral bone marrow lesions in the knee of a non–selective NSAIDS user with metabolic syndrome compared to unchanged subchondral bone marrow lesions in the knee of a selective COX2 inhibitor user with metabolic syndrome.The authors conclude that senescent pre-osteoclasts, by affecting COX2/PGE2 axis in osteoblasts, play a central pathogenic role in metabolic syndrome- induced OA.This paper is very well written and clear. The topic studied is very clinically relevant and has an immediate translational application. The strengths of the paper are the use of two different mouse models and the longitudinal analysis of the human osteoarthritis initiative cohort dataset.There are, however, issues that need to be addressed.Major issues:– Mouse models:– The role of subchondral bone changes is osteoarthritis progression is not new. Others have shown that in OA subchondral bone thickness is increased while trabecular bone thickness is reduced. Overall, previous models have shown increased subchondral bone remodeling with increase in osteoclasts. The changes in subchondral bone are associated with reduced osteochondral integrity and disruption of the barrier between the intra-articular and sub compartmental joint which causes vascular and nerve invasion and endochondral ossification.In this paper, however, the author report the new finding that trabecular bone is increased in the subchondral space and the osteoclast number is decreased in both mouse models of OA- associated metabolic syndrome.It remains unclear how those changes, not previously reported in subchondral bone, are pathogenic in the progression of OA and not just an associated phenomenon. To prove that senescent osteoclasts are the pathogenic culprit of the progression of metabolic syndrome-induced OA, the authors should measure with microCT, the subchondral bone changes and the OA progression (OARSI score) in mice with deletion of senescent osteoclast progenitors fed a high fat diet for 1 month (the mice shown in Figure 4 A), compared to the appropriate controls. If deletion of senescent pre osteoclasts in subchondral bone prevents the increase of subchondral bone thickness in mice fed a high fat diet and the increase of the HFD-induced increase in the OARSI score, this would be a compelling evidence that they could be the culprit of those changes. It would be crucial to use a short term high fat diet as the authors show in Figure 3b that the senescent cells are only present in subchondral bone and not in the articular space with this HFD duration.

We appreciate the valuable comments and constructive suggestion. As mentioned by the reviewer, MetS-associated OA mice used in this study (high fat diet and STR/Ort mice) have distinct subchondral bone phenotype from post-traumatic OA (PTOA) mice, which have been much more intensively investigated in the field. Unlike increased subchondral bone remodeling (increased osteoclast activity) at early-stage OA in the PTOA mice (ACLT and DMM mice), there is a rapid and persistent increase in joint subchondral plate and trabecular thickness during the progression of the MetS-OA mice.

As suggested, we have examined whether senescent osteoclasts are the pathogenic culprit of the progression of MetS-OA by conducting additional in vivo experiment using our established conditional *RANK-Cre; p16 ^flox/flox^* mice (named “p16^cKO^” thereafter). In this mouse line, a key senescence gene *p16^INK4a^* is deleted in RANK^+^ cells. *RANK-Cre* (kindly provided by Dr. Yasuhiro Kobayashi at Matsumoto Dental University, Japan) targets osteoclast lineage cells (*Nat Med.* 2012; 18: 405–412; *Nat Med.* 2016; 22: 1203–1205). In the past several months, we have conducted a systemic characterization on the changes in knee joint subchondral bone and cartilage in the p16^cKO^ mice and the *p16^flox/loxf^* littermate control mice (named “WT” thereafter), especially under HFD challenge for 1 month. Consistent with Figure 3, D-F, SA-βGal+ cells accumulated only in subchondral bone marrow in WT mice with 1-month HFD treatment, and this increase in subchondral SA-βGal+ cells was greatly dampened in the p16^cKO^ mice relative to WT mice (Figure 5—figure supplement 7). The results suggest that deletion of *p16^INK4a^* in RANK^+^ cells efficiently prevents/blocks subchondral cellular senescence. Importantly, microCT analyses showed that tibial subchondral BV/TV ratio, SBP.Th, and Tb.Th were all higher in WT mice fed a HFD (vs. CHD) (Figure 5, A-D); however, these subchondral bone alterations induced by 1-month HFD treatment were not significant in p16^cKO^ mice. Moreover, the reduction in the number of bone surface osteoclasts (Figure 5, E and F) and the increase in the osteoblast clusters in bone marrow (Figure 5, G and H) induced by HFD were both alleviated in p16^cKO^ mice. Therefore, deletion of senescent preosteoclasts in subchondral bone indeed prevents the increase of subchondral thickness induced by a short-term (1 month) HFD treatment.

We also assessed whether deletion of *p16^INK4a^* in preosteoclasts could ultimately lead to attenuated cartilage degeneration induced by HFD using the p16^cKO^ mice. To reach the purpose, the mice have to be fed HFD for a much longer period of time than 1 month to see cartilage degeneration in our mouse model because we found that the MetS-OA mice started to show subchondral thickening phenotype as early as 2 weeks after HFD treatment, but cartilage degeneration did not occur until 3-5 months after HFD treatment (Figure 1). Because of this reason, we fed the p16^cKO^ mice and WT littermate mice with HFD for 5 months. Our data show that HFD-challenged WT mice exhibited obvious proteoglycan loss in the joint cartilage and an increased OARSI score, which were not observed in p16^cKO^ mice (Figure 5, I and J). Consistently, HFD-induced increase in the percentage of MMP13^+^ chondrocytes, another feature of articular cartilage degeneration, was dramatically reduced in p16^cKO^ mice relative to WT mice (Figure 5, K and L). Therefore, blockage of preosteoclast senescence led to a greatly attenuated cartilage degeneration induced by HFD. Of note, *p16^INK4a^* is specifically deleted in osteoclast lineage in bone in the conditional p16^cKO^ mice, and chondrocytes in cartilage were not targeted. Moreover, we did not detect any senescent cells in cartilage in our MetS-OA mouse model until 5 months HFD treatment (Figure 3). Thus, the attenuated cartilage degeneration in the p16^cKO^ mice should be primarily through blockage of subchondral preosteoclast senescence. Collectively, the results from the new panel of experiments suggest that preosteoclast senescence drives pathological increase in subchondral bone volume and subchondral plate thickness, eventually leading to cartilage degeneration.

We agree with the reviewer that the mechanisms of how increases in subchondral bone volume and subchondral plate/trabecular thickness lead to cartilage degeneration were not explored in this study. In fact, the mechanisms by which subchondral bone alteration contribute to OA development/progression is a long-standing question in the OA field in general. There is significant evidence in the literature, however, to suggest that normal subchondral bone structure is essential for the homeostasis of articular cartilage (*Nat. Rev. Rheumatol.* 2011;7:43–49; *Trends Pharmacol. Sci.* 2014;35:227–236) and that the incremental increase of either subchondral bone/plate thickness or subchondral plate stiffness modulus subsequently increases stress and creates uneven stress distributions in the overlaying articular cartilage for its degradation (*Osteoarthritis Cartilage* 2019. 27: 392-400; *Nat Commun*. 2021. 12: 1706). Moreover, strategies reducing/ normalizing subchondral bone alterations during the early stage of OA effectively prevented cartilage degeneration (*JCI Insight.* 2020. 5:e135446; *Nat Med* 2013;19:704-712). Significant ongoing effort in the lab is focused on the functional relationship between subchondral bone and articular cartilage in the MetS-OA models. The discussion on this point has also been added in the revised manuscript (Line 425-442).

– Also the protective effect of Celecoxib in the experiment in Figure 6 does not clearly prove an effect of Cox2 derived from osteoblasts as the beneficial effect of this drug on OA could be mediated by other cell types.

This is an intriguing question. We wished to provide evidence to prove that the beneficial effect of Celecoxib is due to osteoblast/osteocyte-specific COX2 inhibition using conditional *Osteocalcin-Cre; COX2^flox/flox^* or *Dmp1-Cre; COX2^flox/flox^* mouse lines. However, *COX2^flox/flox^* mice were not obtained until this month. We are now breeding this mouse strain with *Osteocalcin-Cre* to generate mice with conditional COX2 deletion in osteoblasts. We look forward to the exciting results from the mice, but it typically takes over a year to generate/validate the mouse line, apply HFD treatment, and conduct analyses of the joint phenotype. Instead, in this present study, we have carefully re-calculated the number of COX2^+^ cells at both articular cartilage and subchondral bone regions in mice at different time points after HFD treatment. Our results showed that there were very few COX2+ cells at cartilage in mice fed HFD for shorter periods (0.5, 1, 3, and 4 months) relative to mice fed CHD, whereas there was a much higher number of COX2^+^ cell number on bone surface (osteoblasts) and in mineralized bone (osteocytes) at these time points (Figure 8, A, C, and D). A higher number of COX2+ chondrocytes in cartilage was found in mice only after 5 months of HFD treatment (Figure 8, A and B), during which severe cartilage degeneration occurs. Therefore, the majority of COX2^+^ cells were primarily subchondral osteoblasts and osteocytes during the progression of MetS-OA, and the major cellular target of celecoxib treatment should be subchondral bone osteoblast lineage cells but not cartilage cells. Discussion on this point has also been added in our revised manuscript (Line 491-494).

Histology:– Figure 2: The authors indicate that the increase in subchondral bone is due exclusively secondary to increased osteoblasts, however the osteoclasts are also reduced on the bone surface. This reduction could be also causing increased bone mass in the subchondral bone. This aspect should be discussed.

This is an important point, and we fully agree with this reviewer that both increased osteoblastic bone formation and decreased osteoclastic bone resorption contribute to the increases in subchondral plate thickening and trabecular bone volume. We have included three new figures (Figure 6—figure supplement 8-10) to demonstrate the effects of preosteoclast senescence and the corresponding SASP on osteoclast differentiation. Our new results suggest that the senescent preosteoclasts, on one hand, have declined differentiation capacity toward mature osteoclasts; on the other hand, secrete SASP factors to inhibit the differentiation of non-senescent osteoclast precursors to osteoclasts in a paracrine manner. The new results have been describe in the Result section (Line 278-307). Discussion on the involvement of reduced osteoclast bone resorption in this process has been added in the revised Discussion section (Line 465-470).

– Figure 2: the number of osteoblasts /bone surface should also be calculated to prove that they are increased. Alternatively, measurements of dynamic indices of bone histomorphometry should be reported (i.e. quantification of bone formation after calcein injection).

As suggested, we have re-evaluated the number of osteoblasts in subchondral bone. We found that, unlike subchondral bone surface localization of the osteocalcin (OCN)^+^ osteoblasts in chow diet control mice, OCN^+^ osteoblasts accumulate and form clusters within marrow cavity in HFD challenged mice. Osteoblast number per bone marrow area (OCN^+^ N/BM.Ar) significantly increased but OCN^+^ osteoblasts per bone surface (OCN^+^ N/BS) remained unchanged in subchondral bone of HFD-challenged mice relative to CHD control mice (Figure 2, A-C). The results suggest that osteoblasts are located on subchondral bone surface in normal healthy joints but aberrantly accumulate in bone marrow cavity in the joints of MetS-OA mice. The formation of aberrant osteoblast clusters within the bone marrow may represent a unique subchondral bone feature of MetS-OA, contributing to the rapid development of subchondral sclerosis and increased subchondral bone mass. The point has been included in the revised Discussion (Line 475-481).

Human longitudinal analysis:– In supplementary table 1 it looks like that OA is less severe in patients with metabolic syndrome (based on KL grade) compared to matched patients without metabolic syndrome. This is the opposite of what it should be expected, as the premise of this research is that OA is more severe in patients with metabolic syndrome. The authors should discuss this contradictory finding.

We appreciate this very constructive comment of the reviewer. We have responded to this comment in two different sections:

1) "OA is less severe in participants with metabolic syndrome (based on KL grade) compared to matched participants without metabolic syndrome": We have re-assessed our propensity score (PS) matching method. A high body-mass index (BMI) is a well-known risk factor for OA (*International Journal of Obesity* 2001; 25: 622-627). On the other hand, while high BMI itself is not included in the definition of MetS (as defined by IDF [*Diabetic Medicine* 2006; 23: 469-480]), it is strongly correlated with abdominal (i.e., central) obesity, which is among MetS criteria. Furthermore, BMI is both associated with knee OA and MetS and can confound the relationship between MetS and OA. Thus, BMI has to be included in the PS-matching model. Since, by MetS definition, all participants with MetS had abdominal obesity (and therefore had high BMIs), PS-matched participants without MetS (MetS^–^) also had high BMIs (see Author response table 1). To address this comment, we have speculated that, given the inclusion of BMI in the PS-matching, MetS^–^ participants may have a relative high BMI after PS-matching and consequently have higher radiographic KL-grades compared to matched MetS^+^ participants. Therefore, we tested this hypothesis with matching the participants once with BMI in the matching model and once without it. We observed that the exclusion of BMI from PS-matching resulted in MetS^+^ participants with higher KL grades compared to PS-matched MetS^–^ participants (Marked as blue in Author response table 1).

**Author response table 1. sa2table1:** Result of PS-matching MetS^+^ and MetS- participants once with BMI inclusion in the PS-matching covariates and once without including BMI. With BMI in the matching model, MetS^+^ participants have lower mean KL grades, and without its inclusion, MetS^+^ participants have higher mean KL grades.

	All OAI subjects		Matched subjects without BMI in matching		Matched subjects with BMI in matching				
MetS^–^	MetS^+^		MetS^–^	MetS^+^		MetS^–^	MetS^+^	
	7459	1810	SMD	1803	1803	SMD	N: 1800	N: 1800	SMD
Variables included in the PS matching model
Age (year) [mean (SD)]	60.38 (9.09)	64.94 (8.49)	0.52	65.22 (8.95)	64.90 (8.48)	0.04	65.12 (8.47)	64.87 (8.46)	0.03
Sex, Female, N (%)	4426 (59.3)	985 (54.4)	0.10	1043 (57.8)	985 (54.6)	0.07	1014 (56.3)	983 (54.6)	0.04
Non-white race [N (%)]	1468 (19.7)	494 (27.3)	0.18	439 (24.4)	487 (27.0)	0.06	481 (26.8)	488 (27.1)	0.01
BMI (kg/m^2^) [mean (SD)]	28.23 (4.78)	30.71 (4.43)	0.54	28.09 (4.52)	30.72 (4.43)	0.59	30.51 (4.53)	30.69 (4.43)	0.04
Smoking, current smoker [N (%)]	517 (6.9)	120 (6.6)	0.01	113 (6.3)	120 (6.7)	0.02	107 (5.9)	120 (6.7)	0.03
Alcohol use, ≥1/week [N (%)]	3250 (43.6)	698 (38.6)	0.10	689 (38.2)	698 (38.7)	0.01	733 (40.7)	696 (38.7)	0.04
PASE score [mean (SD)]	165.42 (83.48)	141.00 (76.02)	0.31	137.63 (72.17)	141.25 (76.03)	0.05	143.72 (78.12)	141.42 (75.89)	0.03
Variables not included in the PS matching model
KL grade, N (%)			0.21			0.12			0.11
Grade 0	2875 (38.5)	540 (29.8)		634 (35.2)	538 (29.8)		511 (28.4)	539 (29.9)	
Grade 1	1318 (17.7)	345 (19.1)		317 (17.6)	342 (19.0)		283 (15.7)	342 (19.0)	
Grade 2	2019 (27.1)	515 (28.5)		496 (27.5)	514 (28.5)		549 (30.5)	511 (28.4)	
Grade 3	1005 (13.5)	332 (18.3)		296 (16.4)	331 (18.4)		375 (20.8)	331 (18.4)	
Grade 4	242 (3.2)	78 (4.3)		60 (3.3)	78 (4.3)		82 (4.6)	77 (4.3)	
Medial JSN score, N (%)			0.22			0.17			0.04
Grade 0	4625 (65.9)	937 (55.9)		1068 (63.4)	933 (55.8)		959 (57.1)	936 (56.0)	
Grade 1	1503 (21.4)	431 (25.7)		379 (22.5)	430 (25.7)		409 (24.4)	427 (25.6)	
Grade 2	742 (10.6)	255 (15.2)		205 (12.2)	254 (15.2)		265 (15.8)	254 (15.2)	
Grade 3	148 (2.1)	54 (3.2)		32 (1.9)	54 (3.2)		46 (2.7)	54 (3.2)	
WOMAC pain score (mean (SD))	2.35 (3.28)	2.96 (3.56)	0.18	2.43 (3.27)	2.96 (3.56)	0.16	2.75 (3.62)	2.95 (3.56)	0.06
Cardio/Cerebrovascular diseases, N (%)	239 (3.3)	196 (11.4)	0.31	79 (4.5)	196 (11.4)	0.26	82 (4.7)	192 (11.2)	0.24
Hypertension [N (%)]	2704 (36.3)	1734 (95.8)	1.62	750 (41.6)	1728 (95.8)	1.44	829 (46.1)	1724 (95.8)	1.31
Diabetes Mellitus [N (%)]	86 (1.2)	646 (36.7)	1.02	18 (1.0)	641 (36.5)	1.02	13 (0.7)	636 (36.3)	1.03
Dyslipidemia [N (%)]	1010 (13.5)	1599 (88.3)	2.26	257 (14.3)	1594 (88.4)	2.21	272 (15.1)	1591 (88.4)	2.16
Abdominal obesity [N (%)]	6293 (84.6)	1810 (100.0)	0.60	1535 (85.4)	1803 (100.0)	0.58	1659 (92.6)	1800 (100.0)	0.40

2) Modifying the subject selection criteria to assess the primary role of pathological subchondral bone structural alterations in the MetS-OA: Following this comment, your other comment (#4), and also similar comments from other reviewers (e.g., Reviewer C, comment #1), to assess the role of subchondral bone structural changes and to better similarize the human participants and mice models of OA design, we tailored our selection criteria. We compared KOA progression in participants with MetS-OA without a history of knee trauma (to exclude PTOA cases) versus participants with PTOA without MetS (to exclude MetS-OA cases). These selection criteria helped us delineate the unique pathophysiology (e.g., prominent and rapid subchondral bone marrow changes) in the MetS-OA phenotype. As we have shown in the animal model, while both PTOA and MetS-OA have degrees of cartilage degradation, prominent and rapid subchondral bone change is a feature of MetS-OA. We gathered data of OAI participants with MRI reads for baseline and 24-month follow-up visits and carefully selected and matched participants. Following your constructive comments, we added KL and medial JSN grades to the PS-matching and adjusted the Cox models for these variables to address the inhomogeneity in the baseline knee OA status and grade. (Supplementary Flowchart 1 and Supplementary Table 1). We compared both standard knee OA outcomes (including JSN progression) and subchondral bone marrow lesion (BML) worsening between these two groups of MetS-OA^+^(PTOA^–^) versus PTOA^+^(MetS-OA^–^). For the assessment of BMLs, we used MRI Osteoarthritis Knee Score (MOAKS) measures, which have been shown to have excellent reliability (*Osteoarthritis and cartilage* 2011; 19: 990-1002; *Arthritis Rheumatol* 2016; 68: 2422-31). MOAKS is a validated and the most commonly used semi-quantitative scoring tool to assess longitudinal change in subchondral BMLs within knee subregions in terms of time points (*Osteoarthritis and cartilage* 2011; 19: 990-1002; *Arthritis Rheumatol* 2016; 68: 2422-31). While we observed that the risk of knee OA radiographic progression in participants with MetS-OA is similar to participants with PTOA during the follow-up period, we found that participants with MetS-OA have more odds of subchondral bone marrow lesion (BML) worsening compared to participants with PTOA (even though they had similar risk of radiographic knee OA progression). This finding was evident because of the higher odds of worsening in the number of knee joint subregions with subchondral BMLs (Table 1). The changes have been reflected in the revised Methods section (Line 552-570 and Line 576-579), Results section (Line 92-119), Table 1, Discussion section (Line 409-411 and Line 501-506), Supplementary material section (Line 116-122, Line 138-164 and Line 181-184, Supplementary Table 1, Supplementary Flowchart 1).

– In the longitudinal analysis of table 1 and 2 more details are needed. The detailed data for the matched patients should be reported in addition to the average length of follow up. In addition, the detailed analysis of the OA scores should be reported and not just the Hazard ratio of progression. Those data would give a better idea on the degree of progression in every group (i.e patient with metabolic syndrome vs matched patients without metabolic syndrome; patients with metabolic syndrome chronically using Cox2 inhibitors vs matched patients not using this drug).

As requested, we have added details of the analysis, the follow-up duration, and the number of events to the manuscript text and tables (see changed Table 1 and 2 in the manuscript). In all BML assessments, we used 24-month worsening in BML MOAKS measures between baseline and 24-month follow-up MRI assessments. Therefore, the follow-up duration for BML assessments was 24 months. Regarding the comparison of knee OA standard outcomes between participants with MetS-OA and PTOA, we have added to the Results section (Line 107) and also footnote of Table 1. Regarding the comparisons of knee OA standard outcomes between COX2 inhibitor users and non-selective NSAID users, we have added to the Results section (Line 370-371) and also footnote of Table 2.

– Based on the data presented in table 1 and 2 metabolic syndrome does not affect the incidence of OA, but worsens the progression of OA. Therefore, in the abstract, the sentence in line 5 should be changed. As reported above the data in table 1 suggest the opposite, as patients without metabolic syndrome have worsen KL score at baseline (grade 2-3-4).

We have changed the mentioned sentence in the Abstract.

– The data presented in the longitudinal studies in humans do not prove the role of subchondral bone thickening and subchondral Cox2 activation on the progression of osteoarthritis. The effects of celecoxib could be due to Cox2 inhibitions in other cells types. The authors present the MRI imaging of just 1 patient where they show subchondral bone changes, however these data, although suggestive, are not sufficient to prove a clear pathogenic mechanism. If possible the data on the subchondral bone in these patients should be reported.

Following your precise and relevant comment, we assessed subchondral BML changes between COX2 inhibitor users and non-selective NSAID users using validated MOAKS measures (*Osteoarthritis and cartilage* 2011; 19: 990-1002; *Arthritis Rheumatol* 2016; 68: 2422-31). Of a total of 94 (47:47) COX2 inhibitor users: non-selective NSAID users with MetS-OA in the Knee OA progression analysis, 88 (44:44) had available baseline and follow-up MRIs. A musculoskeletal radiologist with 12 years of experience read and scored these images according to MOAKS measures. We showed that COX2 inhibitor users have lower odds of 24-month worsening in subchondral BMLs than non-selective NSAID users. New results have been added in the revised manuscript text, Results section (Line 381-387), Methods section (Line 551-570) and tables (Table 3). Details of the manuscript change following this, and your first comment has been presented earlier in response to Comment #1.

Minor issues:– Abstract line 9. Change “undergo” to “undergoing”– Page 12 line 219: the experiment indicates that senescent preosteoclasts secrete SASP factors, not that they are involved in OA pathogenesis. This sentence should be changed– Please add the number of mice used in the experiments for Figure 3, supplementary Figures 3, 4, 5, 7– The authors should indicate the source of OxLDL in the methods

The errors/typos have been changed. The number of mice in all figures have been added in Figure Legends. The source of oxLDL has been added in the Methods section (Supplemental material Line 48).

Response to comments from Reviewer C:This is a highly interesting paper that provides important insights into metabolic syndrome (MetS) associated OA. The authors suggest that in MetS OA subchondral bone changes (as induced by senescent pre-osteoclasts and their specific secretory phenotype) precede cartilage damage and that selective inhibition of COX-2 may have beneficial effects in patients with MetS OA that got beyond the effects seen in non-MetS patients. Generally, this is a very well performed study that may contribute important novel aspects to the field. I have three major comments– The human data as obtained from the OAI dataset are not completely convincing. This is because they largely focus in JSN as a readout for progression, which actually reflects cartilage loss rather than subchondral bone thickness. If the hypothesis of the authors holds true, differences in subchondral bone thickness should be visible particularly in early OA patients with MetS (vs non- MetS patients) and it should be possible to retrieve those data. The same is true for the COX-2 data. Also here, the authors may want to show differences in subchondral bone thickness between (part early) OA patients with/without MetS that had taken COX-2 inhibitors.

We appreciate the constructive comment of the reviewer. Following this comment and similar comments of other reviewers (Reviewer B, comments #1 & #4), we tailored our selection criteria to assess the role of subchondral bone structural changes and to better similarize the human participants and mice models of OA design, we tailored our selection criteria. We compared KOA progression in participants with MetS-OA without a history of knee trauma (to exclude PTOA cases) versus participants with PTOA without MetS (to exclude MetS-OA cases). These selection criteria helped us delineate the unique pathophysiology (e.g., prominent and rapid subchondral bone changes) in the MetS-OA phenotype. As we have shown in the animal model, while both PTOA and MetS-OA have degrees of cartilage degradation, prominent and early subchondral bone alteration is a feature of MetS-OA compared to PTOA. We gathered data of OAI participants with MRI reads for baseline and 24-month follow-up visits and carefully selected and matched participants. Following your constructive comments, we compared both standard knee OA outcomes (including JSN progression) and subchondral bone marrow lesion (BML) worsening between these two groups of MetS-OA^+^(PTOA^–^) versus PTOA^+^(MetS-OA^–^). For the assessment of BMLs, we used MRI Osteoarthritis Knee Score (MOAKS) measures, which have been shown to have excellent reliability (*Osteoarthritis and cartilage* 2011; 19: 990-1002; *Arthritis Rheumatol* 2016; 68: 2422-31). MOAKS is a validated and most commonly used semi-quantitative scoring tool to assess longitudinal change in subchondral BMLs within knee subregions in terms of time points (*Osteoarthritis and cartilage* 2011; 19: 990-1002; *Arthritis Rheumatol* 2016; 68: 2422-31). While we observed that the risk of knee OA radiographic progression in participants with MetS-OA is similar participants with PTOA during the follow-up period, we found that participants with MetS-OA have more odds of subchondral bone marrow lesion (BML) changes compared to participants with PTOA (even though they had similar risk of radiographic knee OA progression). This finding was evident as the higher odds of worsening in the number of knee joint subregions with subchondral BMLs. (Table 1) Further, following your precise and relevant comment, we assessed subchondral BML worsening between COX2 inhibitor users and non-selective NSAID users using validated MOAKS measures (*Osteoarthritis and cartilage* 2011; 19: 990-1002; *Arthritis Rheumatol* 2016; 68: 2422-31). Of a total of 94 (47:47) COX2 inhibitor users: non-selective NSAID users with MetS-OA in the Knee OA progression analysis, 88 (44:44) had available baseline and follow-up MRIs. A musculoskeletal radiologist with 12 years of experience read and scored these images according to MOAKS measures. We showed that COX2 inhibitor users have lower odds of 24-month worsening in subchondral BMLs than non-selective NSAID users. Results have been added to the manuscript text and tables (Table 3). The changes have been reflected in the revised Methods section (Line 552-570 and Line 576-579), Results section (Line 92-119), Table 1, Discussion section (Line 409-411 and Line 501-506), Supplementary material section (Line 116-122, Line 138-164 and Line 181-184, Supplementary Table 1, Supplementary Flowchart 1).

– As one experimental point, the authors used cytokine protein arrays to characterize the specific phenotype of the pre-osteoclasts. This is a useful though somewhat limited approach that isn't entirely up-to-date. A more holistic and unbiased approach (e.g. RNAseq) would have been far more informative.

As suggested, we have performed bulk RNA-sequencing of control and senescent preosteoclasts. The RNA-seq analysis revealed 4,056 differentially expressed genes in the senescent vs. control preosteoclasts (p < 0.05). Comparing our data with previously defined aging/senescence-induced genes (ASIGs) from publicly available mouse RNA-seq data sets (Aging Atlas database; KEGG pathway database; GO database; MSigD database) identified a total of 150 ASIGs in the senescent preosteoclasts (vs. control non-senescent preosteoclasts) (Supplemental Figure 10A). Among these ASIGs, 31 genes were upregulated, and 119 genes were downregulated in the senescent preosteoclasts relative to control cells (Supplemental Figure 10B). In the main biological process and molecular function genes, alterations in “Aging,” “Damaged DNA binding,” “chromatin DNA binding,” and “NF-kappaB binding” are notable for their known links to senescence and SASP triggering (Supplemental Figure 10C). Of note, osteoclast differentiation- and bone resorption-associated genes are among the most significantly downregulated genes (Supplemental Figure 11), indicating a diminished osteoclast differentiation capacity of the preosteoclasts after becoming senescent. These results further validate that preosteoclasts become senescent and exhibit SASP under pathological conditions. In addition, significantly downregulated osteoclast differentiation-associated genes in the senescent preosteoclasts, in combination with the results from the in vitro osteoclast differentiation assay (Supplemental Figure 8, A and B), suggest that the senescent preosteoclasts, on one hand, have declined differentiation capacity toward mature osteoclasts; on the other hand, secrete SASP factors to inhibit the differentiation of non-senescent osteoclast precursors to osteoclasts in a paracrine manner. An important future focus in the laboratory is on single-cell RNA-seq, which will give more insight into gene expression profile of individual cells and offer important understanding on the interplay between senescent cells and their neighboring cells in joint subchondral bone in the context of MetS-OA.